# A major locus confers triclabendazole resistance in *Fasciola hepatica* and shows dominant inheritance

Nicola J. Beesley[1][¤]*, Krystyna Cwiklinski[1], Katherine Allen[1], Rebecca C. Hoyle[1], Terry W. Spithill[2], E. James La Course[3], Diana J. L. Williams[1], Steve Paterson[4], Jane E. Hodgkinson[1]*

1 Veterinary Parasitology, Infection, Veterinary and Ecological Sciences, University of Liverpool, Liverpool, United Kingdom, 2 Department of Animal, Plant and Soil Sciences and Centre for AgriBioscience, La Trobe University, Bundoora, Australia, 3 Liverpool School of Tropical Medicine, Liverpool, United Kingdom, 4 Centre for Genomic Research, Infection, Veterinary and Ecological Sciences, University of Liverpool, Liverpool, United Kingdom

¤ Current address: Department of Biological Sciences, University of Chester, Chester, United Kingdom
* nbeesley@chester.ac.uk (NJB); jhodgkin@liverpool.ac.uk (JEH)

**Data Availability Statement:** All data created during this research is openly available. *Fasciola hepatica* genome and assembly are available on WormBase Parasite BioProject PRJEB25283. Other

## Abstract

*Fasciola hepatica* infection is responsible for substantial economic losses in livestock worldwide and poses a threat to human health in endemic areas. The mainstay of control in livestock and the only drug licenced for use in humans is triclabendazole (TCBZ). TCBZ resistance has been reported on every continent and threatens effective control of fasciolosis in many parts of the world. To date, understanding the genetic mechanisms underlying TCBZ resistance has been limited to studies of candidate genes, based on assumptions of their role in drug action. Taking an alternative approach, we combined a genetic cross with whole-genome sequencing to localise a ~3.2Mbp locus within the 1.2Gbp *F. hepatica* genome that confers TCBZ resistance. We validated this locus independently using bulk segregant analysis of *F. hepatica* populations and showed that it is the target of drug selection in the field. We genotyped individual parasites and tracked segregation and reassortment of SNPs to show that TCBZ resistance exhibits Mendelian inheritance and is conferred by a dominant allele. We defined gene content within this locus to pinpoint genes involved in membrane transport, (e.g. ATP-binding cassette family B, ABCB1), transmembrane signalling and signal transduction (e.g. GTP-Ras-adenylyl cyclase and EGF-like protein), DNA/RNA binding and transcriptional regulation (e.g. SANT/Myb-like DNA-binding domain protein) and drug storage and sequestration (e.g. fatty acid binding protein, FABP) as prime candidates for conferring TCBZ resistance. This study constitutes the first experimental cross and genome-wide approach for any heritable trait in *F. hepatica* and is key to understanding the evolution of drug resistance in *Fasciola* spp. to inform deployment of efficacious anthelmintic treatments in the field.

reads are available on the European Nucleotide Archive Project PRJEB50899: (1) Pooled F2 reads: experiment 1 treated animals' accession numbers: ERS10698902, ERS10698904, ERS10698906, ERS10698908, ERS10698910; experiment 1 untreated animals' accession numbers: ERS10698912, ERS10698900, ERS10698914, ERS10698915, ERS10698916; experiment 2 treated animals' accession numbers: ERS10698901, ERS10698903, ERS10698905, ERS10698911; experiment 2 untreated animals' accession numbers: ERS10698907; ERS10698909; ERS10698913. (2) Six individuals of isolate *Fh*LivR4*pop* used for SNP discovery accession numbers: ERS12749101 – ERS12749106. (3) Field Isolate 1 egg reads: pre-treatment samples accession numbers ERS12749095 – ERS12749097 and post-treatment samples accession numbers ERS12749098 – ERS12749100; groups are matched by colour. Genotyping data has been uploaded to the University of Liverpool Research Data Catalogue and is available at https://doi.org/10.17638/datacat. liverpool.ac.uk/1984. Numerical values used to create (1) Fig 1C are included in S1 Table, (2) Fig 2 and Fig 5 are included in S10 Table and S11 Table, and (3) Fig 3 are included in S12 Table.

**Funding:** This work was supported by UK Research and Innovation Biotechnology and Biological Sciences Research Council (UKRI BBSRC): grant numbers BB/I002480/1 (awarded to JEH, SP, DJLW) and BB/P001912/1 (awarded to JEH, SP, DJLW). The funders had no role in study design, data collection and analysis, decision to publish, or preparation of the manuscript.

**Competing interests:** The authors declare that they have no competing interests.

## Author summary

The common liver fluke, *Fasciola hepatica*, causes disease in livestock worldwide and is a zoonosis, resulting in infection in humans in some parts of the world. The main method of treatment in both humans and animals is the drug triclabendazole (TCBZ) because of its activity against both immature and adult parasites. Although resistance to TCBZ is a substantial threat to control of the parasite, we do not know exactly how the drug acts on the parasites or which regions of the genome, or genes, are inherited by parasites that survive TCBZ treatment. Previous studies have focused on analysing genes that are assumed to be involved in drug action. Here, we took an unbiased approach and scanned the whole parasite genome from both experimental and natural infections to identify areas that respond to TCBZ exposure. We identified a small region, equating to just 0.25% of the genome from our experimental infection that is under TCBZ selection. This genomic region was also selected when naturally infected sheep were treated with TCBZ. We found that parasites surviving treatment only needed to inherit one copy of this resistance region, making it a dominant genetic trait. We showed that the resistance region encodes 30 genes and by characterising their function, we have been able to identify several genes that could confer TCBZ resistance in liver fluke. Our findings substantially advance the understanding of how liver fluke have become resistant to TCBZ and pave the way for molecular tests to detect drug resistant parasites and more effectively target treatments in both livestock and humans.

## Introduction

Amongst the helminth infections that pose a substantial risk to livestock and human health worldwide are the liver flukes *Fasciola hepatica* and *F. gigantica*. In livestock their impact can be extensive, reducing productivity through lower meat and milk yields, increasing liver condemnation, causing greater susceptibility to other infections, and as a cause of mortality [1–6]. In humans it is listed as a neglected tropical disease by the World Health Organisation and estimated that between 2.4 and 17 million people are infected with *Fasciola* spp. worldwide [7,8]. Historically, optimal control of fasciolosis has been through treatment with the highly effective anthelmintic, triclabendazole (TCBZ); the drug of choice in livestock (Fasinex, Novartis) and humans (Egaten, Novartis), respectively [9,10]. The rising threat of liver fluke infection driven by a changing climate, alterations in land use, enhanced movement of livestock and the ability to encroach into new territories is compounded by a growing problem of TCBZ resistance in livestock [11–16]. Similarly, there are increasing reports of the failure of TCBZ to effectively treat *Fasciola* spp. infections in humans [17–19].

Genetic linkage approaches offer a powerful means to map anthelmintic resistance loci, with distinct advantages over candidate gene studies, as no prior knowledge of drug mode of action is required [20]. In trematodes, linkage mapping has identified a sulfotransferase (*SmSULT-OR*) as the cause of oxamniquine resistance in Schistosome parasites, and in the process revealed its route of action, mode of inheritance and provided a path for future rational drug design [21]. This has allowed global mapping of oxamniquine resistance alleles in natural populations [22,23]. Similarly, genome-wide approaches screening populations of parasites phenotyped for their sensitivity to praziquantel have implicated a transient receptor potential channel (*Sm.TRPM$_{PZQ}$*) in praziquantel resistance in *Schistosoma mansoni* [24]. There have been similar successes in parasitic nematode species, with population genomic analyses revealing a single genomic quantitative trait locus (QTL) for ivermectin resistance

[25] and monepantel resistance [26] in *Haemonchus contortus*, culminating in the identification of a putative ivermectin resistance gene, *HCON_00155390:cky-1*, a pharyngeal-expressed transcription factor [27].

Whilst the genetic basis of TCBZ resistance has been a focus of many studies the underlying mechanism remains elusive. A number of candidate genes have been proposed, including β-tubulin, P-glycoprotein (Pgp)-linked drug efflux pumps, Flavin mono-oxygenase (FMO), Cytochrome P450 (CYP450), glutathione S-transferase (GST) and fatty acid binding proteins (FABP), as reviewed recently [14,28]. Currently we lack the understanding of whether there is a common mechanism or pathway involved in TCBZ resistance and how TCBZ resistance is inherited, or if the same mechanism is employed by both adult and immature parasites. This inhibits our ability to monitor development of resistance in the field and limits our capacity to effectively deploy anthelmintic drugs to control *Fasciola* spp. infections.

Herein we demonstrate the first genetic cross and subsequent genomic mapping of a phenotypic trait in *Fasciola* spp. [29]. We successfully generated an F2 cross between TCBZ resistant (TCBZ-R, *Fh*LivR1) and TCBZ susceptible (TCBZ-S, *Fh*LivS1) *F. hepatica* parental isolates. Following *in vivo* phenotyping of F2 parasites and subsequent bulk segregant analysis we identified a ~3.2Mbp locus within the *F. hepatica* genome, comprised of 30 genes, that confers TCBZ resistance. Pooled genotyping of *F. hepatica* eggs pre- and post-TCBZ exposure in naturally infected sheep confirmed that this TCBZ resistance locus was also under selection in the field. Genotyping of individual parental, F1 and F2 recombinants, revealed that TCBZ resistance is primarily a single locus trait that shows dominant inheritance.

## Results

### Genetic cross of *Fasciola hepatica* under experimental conditions

Our capacity to maintain the complete life cycle of *F. hepatica* in the laboratory and exploit clonal expansion within the snail means genetic crossing and linkage mapping studies are possible for this parasite. However, conducting a genetic cross with a parasite that has an indirect life cycle, is a hermaphrodite with the capacity to self-fertilise, whilst also being genetically diverse, is particularly challenging. There is a need to control for its complex reproductive biology and demography, which we did here using phenotypically defined clones and genotyping individual F1 from single miracidium infection of snails. Crossing of the *Fh*LivS1 and *Fh*LivR1 parentals yielded batches of metacercariae (n = 42), the majority of which (n = 36) were F1 crosses, based on the presence of at least two microsatellite markers from each parent. In most cases (n = 33), the *Fh*LivR1 maternal parent was the source of eggs from which F1 crosses were derived. In total, F1 metacercariae from 28 snails were used to generate F1 adults *in vivo* and consequently a pool of F2 eggs (Fig 1). To maximise the number of F2 recombinants for *in vivo* phenotyping we a) performed multiple miracidial infection of snails, b) generated pre-mixed pools of F2 metacercariae from multiple snails prior to infection, c) administered a large F2 metacercarial dose of 400 metacercariae per sheep and d) optimised infection recovery rates (total number of adult parasites recovered from untreated control animals as a proportion of total metacercarial dose administered), which were 21.1% and 22.75%, for Experiment 1 and 2, respectively. Importantly, to determine the impact of TCBZ on genome-wide allele frequency the two pools of F2 used to infect sheep within Experiment 1 and Experiment 2 had a common genetic composition. The number of adult flukes recovered from individual sheep pre- and post- treatment, was significantly different (Fig 1; Mann Whitney W = 25; $P$ = 0.0119 (Experiment 1); 0.00794 (Experiment 2)). When considering all animals within a treatment group for each of the two experiments, drug selection resulted in lower numbers of parasites in TCBZ treated animals, a total of 164 and 119 flukes, compared to the 422 and 455 flukes in

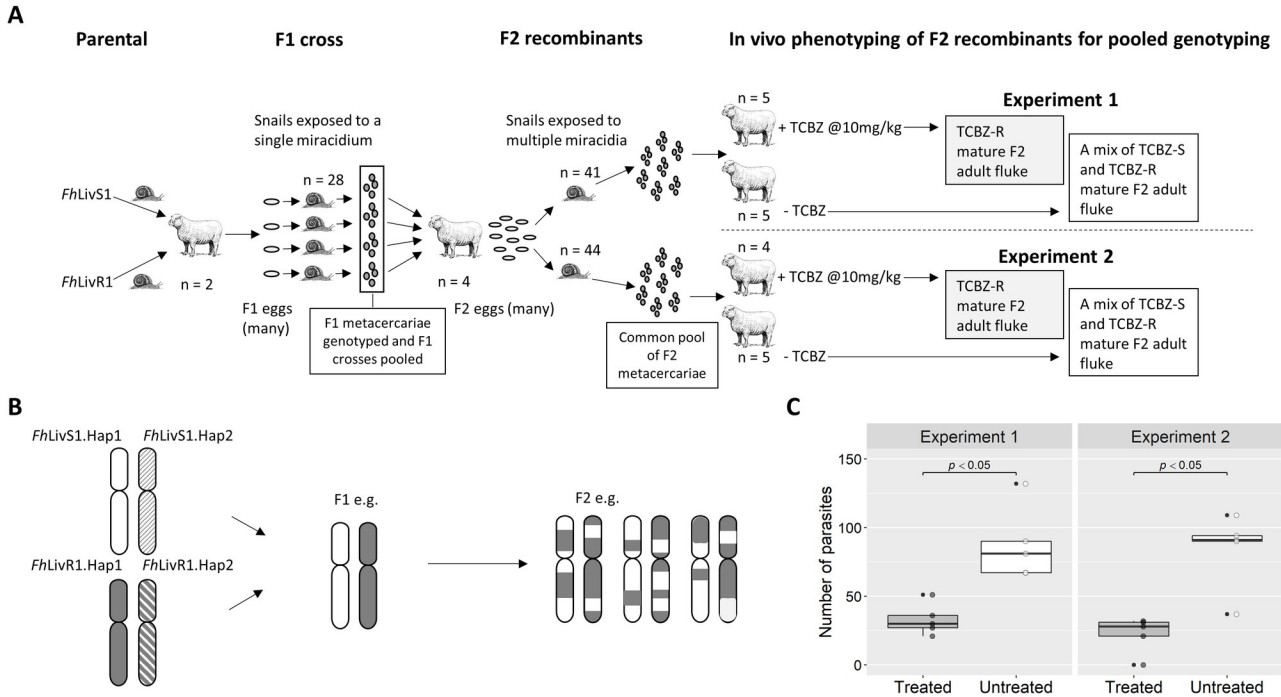

**Fig 1. Experimental overview.** (A) Schematic of the *in vivo* work to produce an F2 cross from *Fh*LivS1 (a clonal population of susceptible parasites) and *Fh*LivR1 (a clonal population of resistant parasites). The parental parasites *Fh*LivS1 and *Fh*LivR1 were produced separately and used to co-infect sheep (n = 2). Some of these parental parasites would cross-fertilise to produce an F1 cross of *Fh*LivS1 and *Fh*LivR1. Eggs were collected from the adult parasites within these sheep. A single miracidium (obtained from these eggs) was used to infect snails (n = 28) and produce clonal F1 populations. The metacercariae were genotyped to ensure they were from an F1 cross and then combined together and used to infect sheep (n = 4). Some of these F1 parasites would cross-fertilise to produce an F2 recombinant population. Eggs were collected from the adult parasites within these sheep. Snails (n = 41 and n = 44 for Experiment 1 and 2, respectively) were exposed to multiple miracidia obtained from these eggs and combined to produce a common pool of F2 metacercariae. For each experiment, two groups of animals were infected with metacercariae from this common pool. Once the infection had reached patency, one group of animals in each experiment was treated with triclabendazole (TCBZ) at a dose of 10mg/kg. At post mortem, those animals which received no treatment had a mixture of triclabendazole susceptible (TCBZ-S) and triclabendazole resistant (TCBZ-R) parasites, whilst those animals that were treated had only TCBZ-R parasites remaining. These parasites were then used for pooled genotyping. (B) A haplotype schematic to show the genetic principle behind the *in vivo* F2 cross. The F1 cross consists of one haplotype from the susceptible parent: *Fh*LivS1 (*Fh*LivS1.Hap1 or *Fh*LivS1.Hap2) and one haplotype from the resistant parent: *Fh*LivR1 (*Fh*LivR1.Hap1 or *Fh*LivR1.Hap2). In the subsequent F2 generation, recombination events take place and the resistant haplotype becomes introgressed amongst the susceptible haplotype producing an F2 recombinant population for study. (C) Plot to show the reduction in the number of F2 parasites recovered from treated animals compared to untreated animals in Experiments 1 and 2. Boxplot indicates the median number of parasites, upper and lower quartiles, and outliers; overlaid points indicate the number of parasites in each animal. In both experiments a significant difference (Mann-Whitney W = 25 $p < 0.05$) is seen in the number of F2 parasites from untreated and treated animals.

untreated hosts, for Experiment 1 and 2, respectively. This represented a reduction of 61% and 74% and an overall recovery rate of 8.2% and 5.95% in treated animals from Experiment 1 and 2 respectively, which constitutes a 2.57- and 3.8-fold reduction in survival of adult parasites in treated hosts compared to untreated controls.

## Genome-wide analysis reveals the same scaffolds under selection in both experimental and naturally occurring recombinants

Genome-wide mapping of genetic determinants for phenotypic traits such as drug resistance relies on a well assembled reference genome. We enhanced our previous *F. hepatica* assembly, increasing scaffold N50 values from 204 Kbp to 1.9 Mb and reducing the number of scaffolds

from 45,354 to 2816, with just 196 scaffolds covering 50% of the genome (Table 1; WormBase ParaSite BioProject PRJEB25283). The completeness of the annotation, as determined by BUSCO, is comparable to that of *Schistosoma mansoni* (WormBase ParaSite 10; BioProject PRJEA36577). Following discovery and filtering, we identified ~9.1M SNPs that segregated between *Fh*LivR1 and *Fh*LivS1 parental clones.

Our approach to mapping loci conferring TCBZ resistance relied on bulk segregant analysis, quantitatively genotyping SNPs in pools of F2 progeny surviving TCBZ treatment and, by comparison with untreated controls, identifying regions of the genome enriched for alleles derived from the resistant parent. In contrast unlinked SNPs (neutral loci) show no difference in allele frequency. We examined the differences in allele frequencies between TCBZ treated (TCBZ+) and TCBZ untreated (TCBZ-) worm pools for each of 9.1M SNPs across the genome, using each sheep as a replicate. The median log-likelihood ratio (LRT) from the generalised linear models (GLM), following bulk segregant analysis from Experiments 1 and 2, is shown in Fig 2A. There was a high degree of concordance between the two experiments, evidenced by over-representation of moving windows of 1000 informative SNPs that independently fell within the 1% highest median LRT in both Experiments 1 and 2 (chi-square test, 1 d.f., 10.952, $p < 0.001$). We identified 6 scaffolds (13, 157, 166, 324, 1853 and 2049) of particular interest because they each had at least 10 moving windows in the top 1% of median LRT in both experiments, suggesting that these were due to a consistent signal of selection within the regions of the genome that they represent. Scaffold 157 showed the greatest evidence of selection (Fig 2A and Table 2).

We then tested whether the same locus was subject to drug selection under natural field conditions. Bulk segregant analysis of naturally occurring *F. hepatica* recombinants under drug selection in the field (Field Isolate 1) demonstrated selection of genes on scaffolds 157 and 1853 (Table 3). The median LRT from the GLM following bulk segregant analysis of eggs pre- and post-TCBZ treatment identified scaffolds under selection; scaffold 157 with 206 out of 615 moving windows and scaffold 1853 with 41 of 104 moving windows in the top 1% (Fig 2B and Table 3). Corroboration between the experimental cross and naturally occurring recombinants, indicating the same two scaffolds (1853 and 157) were under drug selection, supports inheritance of genes on these scaffolds as a means of conferring TCBZ resistance (Fig 2).

**Table 1. *Fasciola hepatica* assembly statistics.**

| Metric | Value |
|---|---|
| Total length of scaffolds | 1.20 Gbp |
| Number of scaffolds | 2816 |
| N50 scaffold length | 1.90 Mbp |
| L50 scaffold count | 196 |
| Number of contigs | 67333 |
| N50 contig length | 39 Kbp |
| L50 contig count | 39050 |
| Total length of gaps | 40 Mbp |
| Average gap length | 620 bp |
| GC content | 42% |
| Number of coding gene models | 9709 |
| Average coding gene model length | 46 Kbp |
| Average number of exons per gene model | 8.1 |
| Average protein model length (number of amino acids) | 570 |
| BUSCO transcripts (complete / fragmented / missing) | 84% / 8% / 8% |

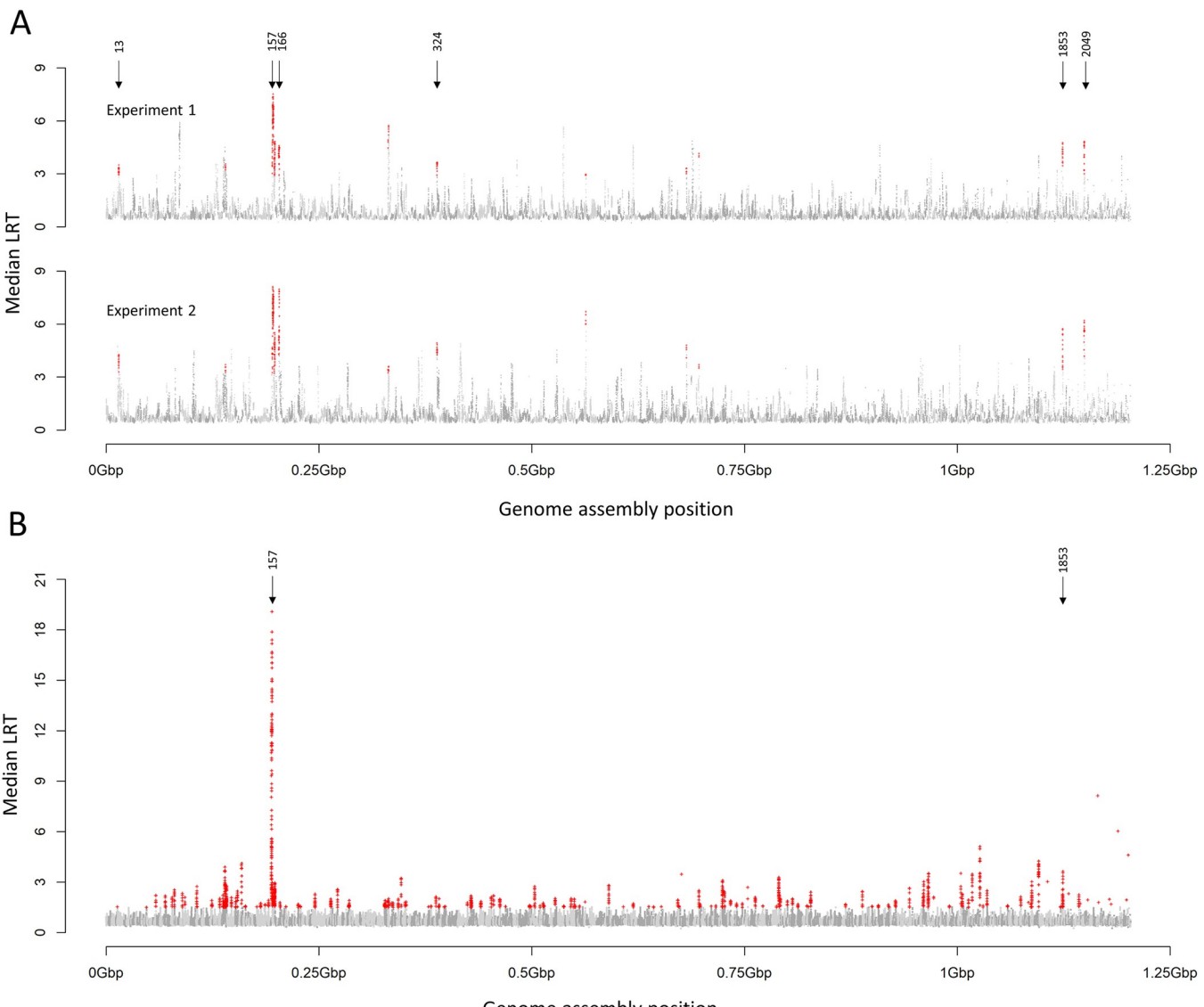

**Fig 2. Genome scan for regions associated with resistance to triclabendazole.** Data show the median likelihood ratio test (LRT) statistic from generalised linear models within moving windows of 1000 informative SNPs. Scaffolds are represented in alternating dark grey and light grey to allow visualisation. Scaffold order on the x-axis is arbitrary and does not imply physical proximity. (A) Results of the two replicate crossing experiments. Position of scaffolds under greatest selection (13, 157, 166, 324, 1853 and 2049) is indicated by arrows. Red crosses indicate where the median LRT appears in the top 1% quantile in both experiments. (B) Results from Field Isolate 1. Position of scaffold 157, under greatest selection, and scaffold 1853 are indicated by arrows. Red crosses indicate where the median LRT appears in the top 1% quantile.

## Triclabendazole resistance is conferred by a single genomic locus

Given that our *F. hepatica* genome is comprised of ~2800 scaffolds we investigated whether the six mapped scaffolds are linked, by genotyping individual parasites and performing linkage analysis. We genotyped 485 x F2 TCBZ- parasites (S3 Table) with a subset of 48 SNPs derived from each of the six mapped scaffolds and 16 SNPs from neutral (not under selection) scaffolds of comparable size (S4 Table). Linkage is shown by a heat map using |D'| values (Fig 3); pairs of SNPs on scaffolds under selection had high |D'| values (median = 0.942, range = 0.264 to 1) which were usually significant, whilst pairs of SNPs that included neutral scaffolds generally

**Table 2. Scaffolds showing evidence of selection in the experimental cross.**

| Scaffold identity | Size of scaffold (base pairs) | No. of moving windows that appear in the 1% quantile[a] | Total no. of windows[a] |
|---|---|---|---|
| scaffold10x_157_pilon | 4023384 | 146 | 326 |
| scaffold10x_166_pilon | 3680061 | 29 | 170 |
| scaffold10x_13_pilon | 7591173 | 17 | 524 |
| scaffold10x_1853_pilon | 874934 | 17 | 39 |
| scaffold10x_2049_pilon | 967921 | 18 | 69 |
| scaffold10x_324_pilon | 2879379 | 15 | 239 |

[a] the median likelihood ratio test statistic from generalised linear models within moving windows of 1000 informative SNPs. If the median LRT for a window is in the top 1% quantile for each of the two replicate experiments it is counted. Those scaffolds with the greatest number of moving windows in this 1% quantile are considered to be those under greatest selection.

had low |D'| values (median = 0.199, range = 0.013 to 1) and were typically not significant. Thus, the six scaffolds under selection in our experimental cross were in linkage disequilibrium.

The haplotypes of the scaffolds under selection inferred by PHASE 2.1.1 [30,31] were reordered to minimise recombination events. Our analysis allowed us to place the scaffolds in order relative to one another (1853, 157, 2049, 166, 324 and 13; Fig 3). After removal of duplicate genotypes (i.e. clones that have arisen following clonal expansion in the snail intermediate host) a total of 39 different haplotypes for the six scaffolds under selection, and 98 different haplotypes for the neutral scaffolds were inferred from PHASE 2.1.1 (S5 and S6 Tables). This equated to 136 genotypes, including 113 unique recombinant F2 genotypes. Further analysis of recombinant haplotypes from surviving parasites (i.e. resistant haplotypes) also allowed us to exploit areas of recombination around scaffold 157 and the surrounding scaffolds to finer scale map the region (Fig 4). SNP markers identified that a single genomic locus, including a 0.3Mbp region of scaffold 1853 and a 2.9Mbp region of scaffold 157, was consistently inherited in resistant parasites (S5 Table). *Fasciola hepatica* has a 1.25Gbp genome and this 3.2Mbp locus constitutes just 0.25% of the genome, encoding 30 genes.

## Triclabendazole resistance shows dominant inheritance

Our experimental genetic cross between a TCBZ-R and a -S isolate confirmed that TCBZ resistance is a heritable trait. By SNP genotyping individual parents, F1 parasites, and 249 x F2 TCBZ+ and 485 x F2- parasites (S3 Table) we could track segregation and reassortment of SNPs through the generations to determine the mode of inheritance. We identified parental SNP genotypes; the TCBZ resistant parent was designated *Fh*LivR1.Hap1/ *Fh*LivR1.Hap2 and

**Table 3. Scaffolds showing evidence of selection in Field Isolate 1.**

| Scaffold identity | Size of scaffold (base pairs) | No. of moving windows that appear in the 1% quantile[a] | Total no. of windows[a] |
|---|---|---|---|
| scaffold10x_157_pilon | 4023384 | 206 | 615 |
| scaffold10x_102_pilon | 4923131 | 125 | 511 |
| scaffold10x_851_pilon | 2133737 | 47 | 280 |
| scaffold10x_1853_pilon | 874934 | 41 | 104 |
| scaffold10x_742_pilon | 3094546 | 31 | 487 |
| scaffold10x_362_pilon | 1064730 | 30 | 137 |

[a] the median likelihood ratio test statistic from generalised linear models within moving windows of 1000 informative SNPs.

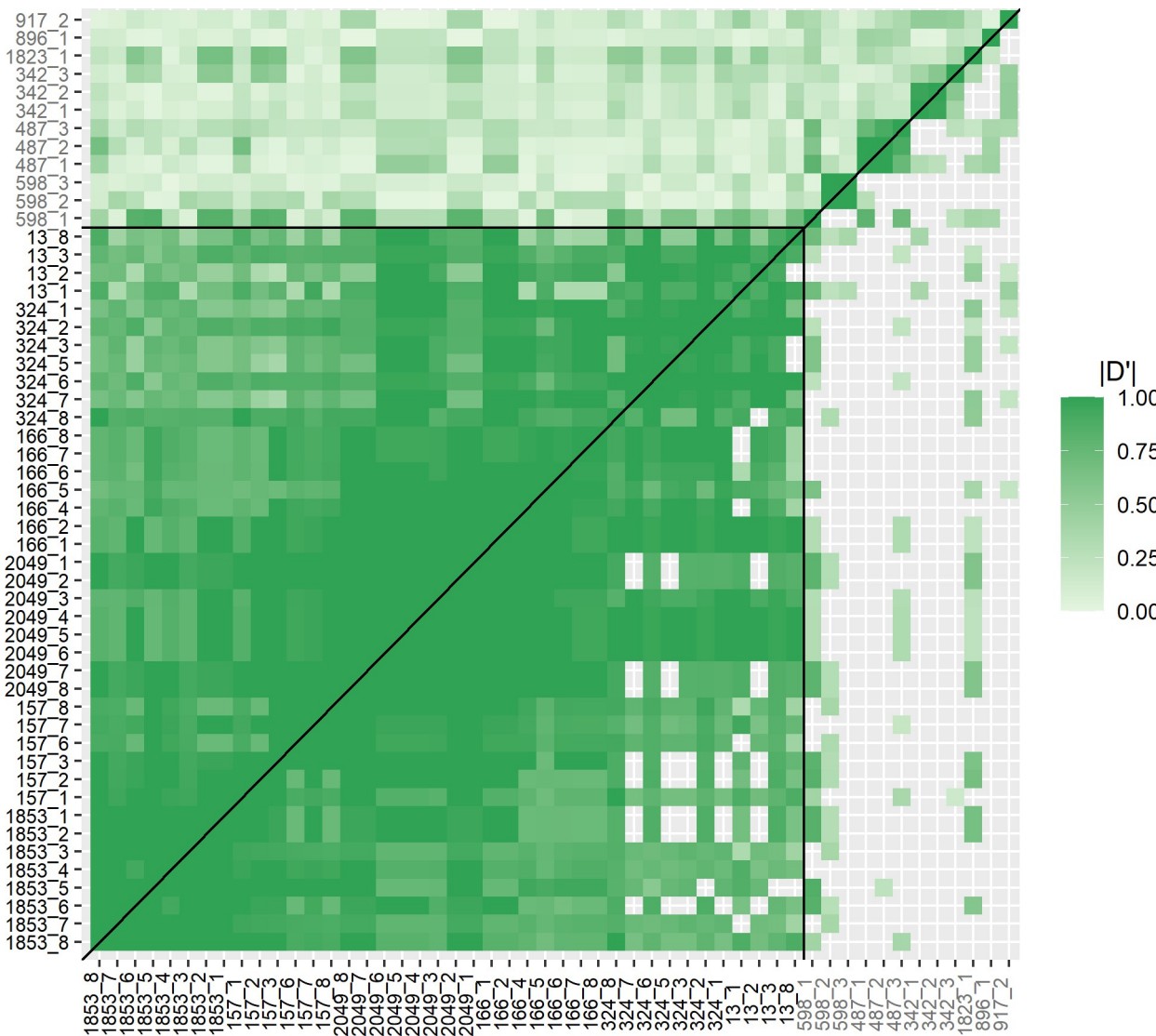

**Fig 3. Heat map (with no clustering or scaling) to show |D'| values between all pairs of loci in untreated F2 parasites.** Loci under selection are enclosed by the black horizontal and vertical lines, with neutral loci outside. Above the diagonal all |D'| values are shown and below the diagonal only |D'| values with significant *q*-values (*q* < 0.05 after false discovery rate correction) are shown. When comparing pairs of loci from scaffolds under selection, high |D'| values indicate that all six scaffolds are in linkage disequilibrium. The |D'| values for the majority of loci pairs containing neutral scaffolds are low or not-significant.

the susceptible parent *Fh*LivS1.Hap1/ *Fh*LivS1.Hap2. It was then possible to assign each F2 parasite as one of 10 different genotypes based on SNP haplotypes. We tested the assumption that TCBZ resistance was a recessive trait, in which case parasites surviving treatment would have two copies of the 'drug resistant' haplotype, or a dominant trait, where only one copy of the resistant haplotype would be sufficient to confer resistance. Resistant parasites from drug treated hosts required only one copy of *Fh*LivR1.Hap2 to survive treatment. Consistent with our finer scale mapping, this was due to the inheritance of the SNPs located on scaffold 157 and 1853 (partial) in parasites that survived TCBZ treatment (Table 4). The parental *Fh*LivR1. Hap1 haplotype did not confer resistance. Thus, we can infer that TCBZ resistance shows dominant inheritance and that the resistant parent used in the experimental cross was heterozygous for the resistance allele.

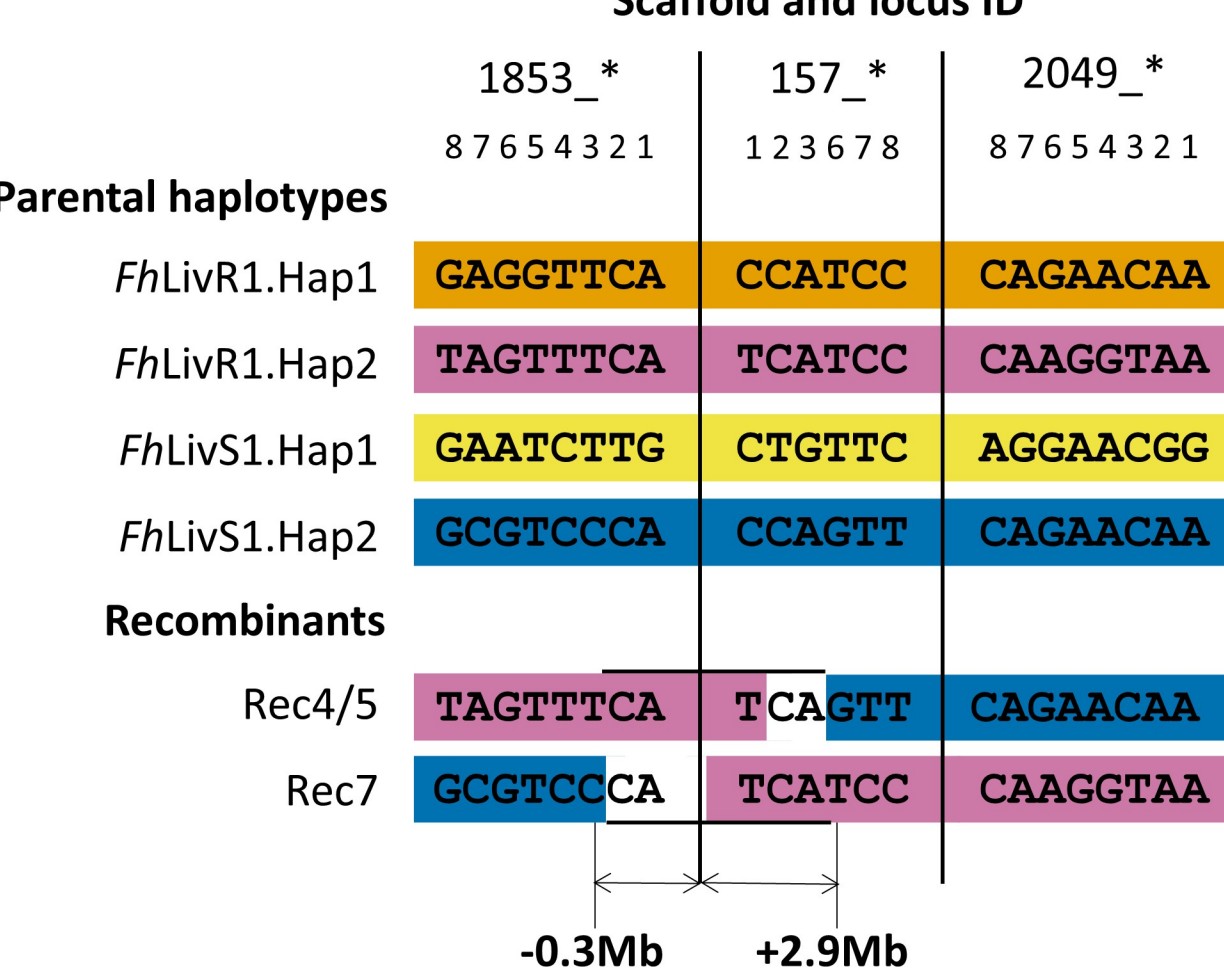

**Fig 4. Schematic to demonstrate finer scale mapping of the genomic region under selection in recombinant F2s compared to parental haplotypes.** Parasites (treated and untreated) were individually genotyped at 36 loci across the six scaffolds under selection. PHASE 2.1.1 [30,31] was used to infer haplotypes from the SNP data and the parental haplotypes (*Fh*LivR1.Hap1; *Fh*LivR1.Hap2; *Fh*LivS1.Hap1; *Fh*LivS2.Hap2) were identified. The figure shows the individual genotypes for the loci on scaffolds 1853, 157 and 2049 (note that even though the sequences are consecutive in the diagram the individual loci are not physically next to each other; the nucleotide position of these loci across each scaffold can be found in S4 Table). Analysis of informative resistant recombinant haplotypes (Rec4/5 and Rec7; S5 Table) found within surviving F2 parasites (i.e. those from treated animals) allowed us to further localise the area needed for a parasite to be resistant. In these recombinants, recombination between SNPs delineates a single genomic locus from 1853_3 to 157_6 (~3.2Mbp; 0.3Mbp region of scaffold 1853 and a 2.9Mbp region of scaffold 157) that was consistently inherited in surviving F2 parasites (S5 Table).

## Characterisation of candidate genes within the triclabendazole resistance locus

We identified 30 candidate genes for TCBZ resistance within the ~3.2Mbp locus based on our annotated genome and by cross referencing with all available gene annotations (Table 5). Any of these 30 genes may confer TCBZ resistance, but the data from Field Isolate 1 highlights a strong signal of selection at the start of scaffold 157, identifying a cluster of genes involved in membrane transport, signal transduction and cell signalling, and DNA/RNA binding and transcriptional regulation (Fig 5, genes 3–10; Table 5). Amongst this cluster of genes are those that have been the focus of previous studies on TCBZ action and/or resistance mechanisms, namely an ADP ribosylation factor (Gene 7: ARF, maker-scaffold10x_157_pilon-snap-gene-0.197; Fig 5 and Table 5), a Ras-related protein (Gene 10: Ras-RP, maker-

**Table 4. Genotyping of individual F2 parasites to determine inheritance of parental SNP haplotypes.**

| Scaffold* | TCBZ treatment | No. of individual parasites with each genotype | | | | | | | | | |
|---|---|---|---|---|---|---|---|---|---|---|---|
| | | *Fh*LivR1.Hap2 + *Fh*LivR1.Hap2 | *Fh*LivR1.Hap1 + *Fh*LivR1.Hap2 | *Fh*LivR1.Hap2 + *Fh*LivS1.Hap1 | *Fh*LivR1.Hap2 + *Fh*LivS1.Hap2 | *Fh*LivR1.Hap1 + *Fh*LivS1.Hap1 | *Fh*LivS1.Hap1 + *Fh*LivS1.Hap2 | *Fh*LivS1.Hap2 + *Fh*LivR1.Hap1 | *Fh*LivS1.Hap2 + *Fh*LivS1.Hap2 | *Fh*LivR1.Hap1 + *Fh*LivR1.Hap1 | *Fh*LivS1.Hap1 + *Fh*LivS1.Hap1 |
| All | - | 8 | 26 | 31 | 37 | 14 | 10 | 43 | 57 | 0 | 0 |
| | + | 10 | 24 | 31 | 47 | 0 | 0 | 0 | 0 | 0 | 0 |
| 1853 | - | 8 | 28 | 46 | 58 | 20 | 26 | 70 | 73 | 1 | 1 |
| | + | 10 | 30 | 50 | 70 | 0 | 0 | 0 | 2 | 0 | 0 |
| 157 | - | 8 | 29 | 40 | 58 | 20 | 23 | 68 | 67 | 5 | 0 |
| | + | 10 | 32 | 40 | 70 | 0 | 0 | 0 | 0 | 0 | 0 |
| 2049 | - | 11 | ND | 32 | 85 | ND | 52 | ND | 151 | ND | 0 |
| | + | 18 | ND | 35 | 81 | ND | 9 | ND | 15 | ND | 0 |
| 166 | - | 11 | 34 | 32 | 47 | 20 | 32 | 78 | 70 | 1 | 0 |
| | + | 18 | 25 | 35 | 56 | 0 | 13 | 13 | 2 | 0 | 0 |
| 324 | - | 11 | 34 | 40 | 41 | 15 | 33 | 71 | 70 | 0 | 0 |
| | + | 18 | 26 | 38 | 52 | 0 | 13 | 13 | 2 | 0 | 0 |
| 13 | - | 11 | 35 | 42 | 40 | 25 | 30 | 64 | 67 | 0 | 3 |
| | + | 18 | 26 | 36 | 50 | 11 | 13 | 2 | 2 | 0 | 0 |

* Scaffold location for each SNP haplotype. ND = not determined due to inability to differentiate between *Fh*LivR1.Hap1 and *Fh*LivS1.Hap2 haplotypes on this scaffold. Shaded region = resistant parasites (those surviving drug treatment in vivo) had to have at least one copy of *Fh*LivR1.Hap2.

scaffold10x_157_pilon-snap-gene-0.182; Fig 5 and Table 5), and an ABCB1 gene (Gene 5: ABCB1, maker-scaffold10x_157_pilon-snap-gene-0.179; Fig 5 and Table 5). Lying within the mapped locus, albeit slightly outside the strongest signal of selection, is another gene that has been the focus of candidate gene studies for TCBZ resistance, a FABP V gene (Gene 17: FABPV, maker-scaffold10x_157_pilon-snap-gene-0.187; Fig 5 and Table 5).

Although several candidate genes fall within the mapped region, it is likely that only one gene within the locus is driving selection. To prioritise amongst gene candidates, we first determined differential expression across life cycle stage-specific transcriptomes (S8 Table). With the exception of the ABCB1 gene (maker-scaffold10x_157_pilon-snap-gene-0.179), two EGF-like proteins (maker-scaffold10x_157_pilon-snap-gene-0.185 and maker-scaffold10x_157_pilon-snap-gene-0.196) and a serine rich protein (maker-scaffold10x_157_pilon-snap-gene-0.188), all the candidate genes were transcribed by the three major *F. hepatica* life cycle stages, namely newly excysted juveniles (NEJ), immature fluke 21 days post infection and adult fluke (with TPM values ranging from 2–510). The most abundantly transcribed genes were the ADP ribosylation factor (maker-scaffold10x_157_pilon-snap-gene-0.197) and an uncharacterised protein (maker-scaffold10x_1853_pilon-snap-gene-0.13), with highest transcript levels present in adult parasites.

We prioritised candidate genes further by interrogating our genomic and genetic data. There was no evidence for difference in copy number variants (CNV) for the ABCB1 gene (maker-scaffold10x_157_pilon-snap-gene-0.179) and most of our prime candidates (genes 3–10; Table 5) were invariant or contained only synonymous mutations within coding regions. Three non-synonymous SNPs that segregated within the experimental crosses were noted, two within the ABCB1 gene (maker-scaffold10x_157_pilon-snap-gene-0.179 gene), $T^{830}A$ and $S^{852}G$, and one within the ADP ribosylation factor (maker-scaffold10x_157_pilon-snap-gene-0.197), $C^{167}Y$ (S9 Table). On initial inspection the $C^{167}Y$ variant was conserved in other TCBZ-R isolates (*Fh*LivR2, *Fh*LivR3, *Fh*LivR4*pop*).

**Table 5. Candidate genes identified following fine scale mapping of scaffolds under selection.**

| Gene no.[1] | Gene id (scaffold id in bold) | Predicted Protein Description[2] | Orthologous genes[3] | | | | |
|---|---|---|---|---|---|---|---|
| | | | *Fasciola hepatica*[4] | *Fasciola gigantica* | *Clonorchis sinensis* | *Opisthorchis viverrini* | *Schistosoma mansoni* |
| 1 | maker-scaffold10x_**1853**_pilon-snap-gene-0.14 | 26S proteasome non-ATPase regulatory subunit 14 | THD25197 | FGIG_04443 | CSKR_111129 | T265_0330 | Smp_213550 |
| 2 | (maker-scaffold10x_**1853**_pilon-snap-gene-0.15) | | | | CSKR_111128 | T265_03304 | |
| 3 | maker-scaffold10x_**1853**_pilon-snap-gene-0.13 | Uncharacterised protein | THD20870 | FGIG_05173 | CSKR_100114 | T265_00015 | Smp_128530 Smp_128520 |
| 4 | maker-scaffold10x_**157**_pilon-snap-gene-0.196 | EGF-like protein | THD24986 | FGIG_08011 | CSKR_110651 CSKR_110646 | T265_12032 T265_14151 | Smp_170980 |
| 5 | maker-scaffold10x_**157**_pilon-snap-gene-0.179 | Putative multidrug resistance protein 1, 2, 3 (P glycoprotein 1, 2, 3); ATP binding cassette subfamily B MDR TAP | THD24985 | FGIG_06362 | ND | ND | Smp_089200 |
| 6 | maker-scaffold10x_**157**_pilon-snap-gene-0.180 | SANT/Myb-like DNA-binding domain protein | THD20256 | FGIG_08588 | ND | ND | Smp_088660 |
| 7 | maker-scaffold10x_**157**_pilon-snap-gene-0.197 | ADP-ribosylation factor 2 | THD20255 | FGIG_08587 | CSKR_110178 | T265_14125 | Smp_088650 |
| 8 | maker-scaffold10x_**157**_pilon-snap-gene-0.181 | RNA-binding protein sym-2/ Heterogeneous nuclear ribonucleoprotein | THD27617 | FGIG_05275 | CSKR_111286 | T265_10181 | ND |
| 9 | maker-scaffold10x_**157**_pilon-snap-gene-0.198 | DNA directed RNA Polymerase I and III (A/C) shared subunit | THD27616 | FGIG_05274 | CSKR_111285 | T265_10182 | Smp_004640 |
| 10 | maker-scaffold10x_**157**_pilon-snap-gene-0.182 | Ras-related protein Rap-1 | THD24364 | FGIG_10598 | CSKR_105623 | T265_00309 | Smp_142450 |
| 11 | maker-scaffold10x_**157**_pilon-snap-gene-0.183 | Receptor protein serine/threonine kinase | THD27619 | FGIG_07463 | CSKR_105620 | T265_12512 | ND |
| 12 | maker-scaffold10x_**157**_pilon-augustus-gene-0.97 | D-amino-acid oxidase/ D-aspartate oxidase | THD27618 | FGIG_07464 | CSKR_105619 | T265_00312 | Smp_170430 |
| 13 | maker-scaffold10x_**157**_pilon-snap-gene-0.184 | Max-like protein X | THD21326 | FGIG_05407 | CSKR_105621 | T265_00311 | Smp_142400 |
| 14 | maker-scaffold10x_**157**_pilon-snap-gene-0.185 | EGF-like protein | THD21325 | FGIG_05408 | CSKR_105622 | T265_00310 | ND |
| 15 | maker-scaffold10x_**157**_pilon-snap-gene-0.186 | Surfeit locus protein 4 | THD25801 | FGIG_06483 | CSKR_114153 | T265_00137 | Smp_174450 |
| 16 | augustus_masked-scaffold10x_**157**_pilon-processed-gene-0.14 | TFIIH basal transcription factor complex helicase XPD subunit | THD25800 | FGIG_06484 | CSKR_114154 | T265_00138 | Smp_199100 |
| 17 | maker-scaffold10x_**157**_pilon-snap-gene-0.187 | Fatty acid binding protein V | THD26047 | FGIG_05100 | CSKR_105127 | T265_00140 | ND |
| 18 | maker-scaffold10x_**157**_pilon-snap-gene-0.200 | Stomatin-2 / SPFH Domain / Band 7 family protein | THD26726 | FGIG_07254 | CSKR_105126 | T265_12460 | Smp_122810 |
| 19 | maker-scaffold10x_**157**_pilon-snap-gene-0.201 | Glycosylphosphatidylinositol (GPI) ethanolamine phosphate transferase 1 | THD23879 | FGIG_05250 | CSKR_102272 CSKR_105203 | T265_07846 T265_07847 T265_14409 | Smp_065130 |
| 20 | maker-scaffold10x_**157**_pilon-pred_gff_StringTie-gene-0.138 | Sugar phosphate exchanger 3 | THD20850 | FGIG_09966 | CSKR_107885 | T265_11304 | ND |
| 21 | maker-scaffold10x_**157**_pilon-snap-gene-0.203 | Ribonuclease 3 | THD20851 | FGIG_09964 | ND | T265_11305 | Smp_142510 |
| 22 | maker-scaffold10x_**157**_pilon-snap-gene-0.188 | Putative serine-rich repeat protein | THD21740 | FGIG_08994 | CSKR_112800 | T265_14787 | ND |
| 23 | maker-scaffold10x_**157**_pilon-snap-gene-0.204 | Putative transferase CAF17, mitochondrial | THD21739 | FGIG_08993 | CSKR_112799 | T265_09217 | Smp_170950 |

(*Continued*)

**Table 5.** (Continued)

| Gene no.[1] | Gene id (scaffold id in bold) | Predicted Protein Description[2] | Orthologous genes[3] | | | | |
|---|---|---|---|---|---|---|---|
| | | | *Fasciola hepatica*[4] | *Fasciola gigantica* | *Clonorchis sinensis* | *Opisthorchis viverrini* | *Schistosoma mansoni* |
| 24 | maker-scaffold10x_**157**_pilon-snap-gene-0.205 | Lamin-1/ Neurofilament protein | THD20852 | FGIG_03874 | CSKR_100679 | T265_05285 | Smp_170930 |
| 25 | maker-scaffold10x_**157**_pilon-snap-gene-0.189 | Gyf domain protein | THD26942 | FGIG_00005 | CSKR_102610* CSKR_100668* | ND | ND |
| 26 | snap_masked-scaffold10x_**157**_pilon-processed-gene-0.72 | Prominin | THD26941 | FGIG_00004 | CSKR_100671 | T265_05281 | Smp_179660 |
| 27 | maker-scaffold10x_**157**_pilon-snap-gene-0.206 | Phospholipid transport protein / CRAL-TRIO / SEC14-like | THD26940 | FGIG_00429 | CSKR_100676 | T265_05283 T265_01975 T265_05284 | Smp_242130 |
| 28 | maker-scaffold10x_**157**_pilon-snap-gene-0.190 | Ubiquitin carboxyl-terminal hydrolase | THD22746 | FGIG_01900 | ND | ND | ND |
| 29 | maker-scaffold10x_**157**_pilon-snap-gene-0.207 | Ubiquitin carboxyl-terminal hydrolase | THD20737 | FGIG_01900 | ND | ND | Smp_128770 |
| 30 | maker-scaffold10x_**157**_pilon-augustus-gene-0.89 | Ubiquitin carboxyl-terminal hydrolase | THD20737 | ND | ND | ND | Smp_152000 |

[1.] Gene number corresponds with Fig 5

[2.] Protein description and function were determined using UniProt Blast, WormBase ParaSite Version 14 Blast, OrthoDB version 9, and InterPro

[3.] Orthologs identified using WormBase ParaSite Version WBPS16 in *Fasciola gigantica* (BioProject PRJNA230515), *Clonorchis sinensis* (BioProject PRJNA386618- Cs-k2), *Opisthorchis viverrini* (BioProject PRJNA222628), and *Schistosoma mansoni* (BioProject PRJEA36577)

[4.] Comparative analysis was carried out against the re-assembled/re-annotated *Fasciola hepatica* genome (BioProject PRJNA179522); ND = not determined

* 1 to many orthologs

## Discussion

### A major locus, that shows dominant inheritance, confers triclabendazole resistance

We have demonstrated that TCBZ resistance exhibits Mendelian inheritance and is conferred by a dominant allele at a single locus. This is the first linkage mapping study for any phenotypic trait in *F. hepatica* and has important implications for our understanding of how drug resistance emerges and spreads in liver fluke populations. A particular strength of our work is the concordance of the classical genetic mapping approach with the field study. We chose to perform the experimental cross with TCBZ-R and -S parasites that were recently isolated from naturally infected sheep in the UK and rendered clonal by laboratory infection of snails [32]. This may explain why we found good agreement between the experimental approach and results from the outbred field populations under natural TCBZ selection. The provenance of both the *FhLivR1* clone and Field Isolate 1 places them around 50 miles from one another in the Northwest of the UK, so perhaps such consistency might be expected. Analysis of further isolates will reveal if this genomic locus underpins TCBZ resistance in more geographically dispersed isolates within the UK and beyond.

Our approach used pooled genotyping, which enhanced the statistical power and precision of the study [20]. Given the complexity of fluke biology it is difficult to know the final number of F2 recombinants used for *in vivo* phenotyping in the experimental cross but, based on our experimental design, we can estimate a minimum of 16 and 12 for resistant parasites (TCBZ+) and 42 and 47 for parasites from untreated animals (TCBZ-), in Experiment 1 and 2, respectively. This is broadly consistent with our observation of 113 unique F2 genotypes. The

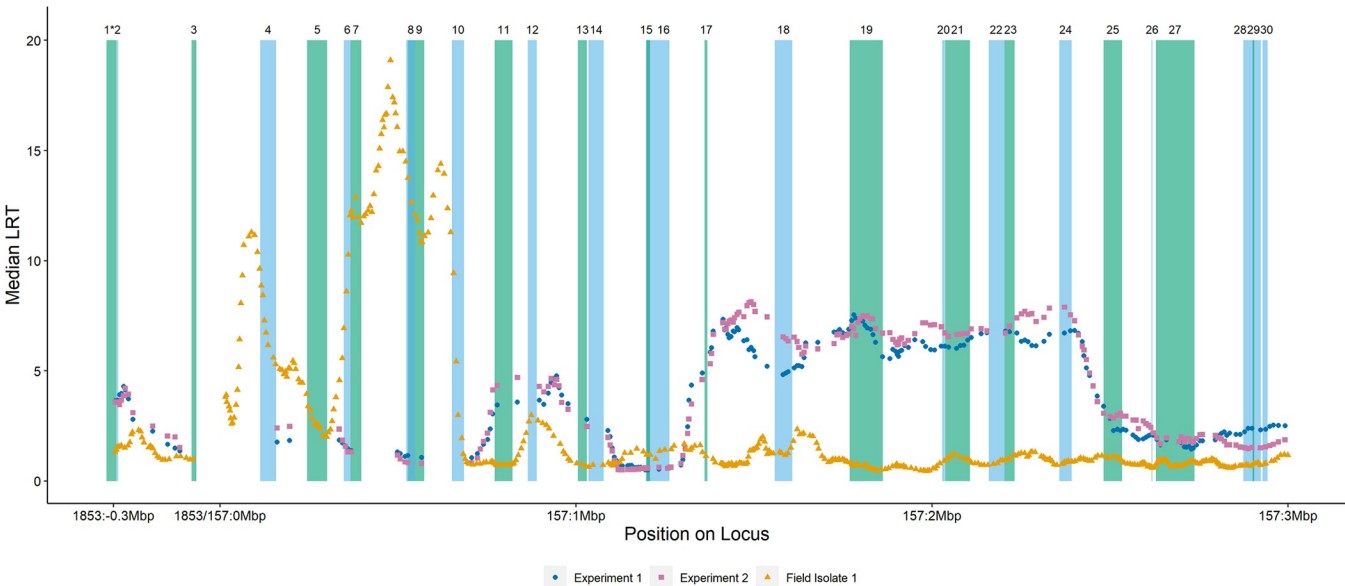

**Fig 5. Median likelihood ratio test (LRT) statistic from generalised linear models within moving windows of 1000 informative SNPs for *in vivo* Experiment 1 and Experiment 2 and Field Isolate 1 are plotted against the position within the 3.2Mbp locus (0.3Mbp region of scaffold 1853 and a 2.9Mbp region of scaffold 157).** Positions of the 30 genes are indicated across the locus and are represented in alternating green and blue colours to allow visualisation. Gene numbering corresponds with Table 5: 1: maker-scaffold10x_1853_pilon-snap-gene-0.15 (26S proteasome non-ATPase regulatory subunit 14; *gene crosses locus boundary); 2: maker-scaffold10x_1853_pilon-snap-gene-0.14 (26S proteasome non-ATPase regulatory subunit 14); 3: maker-scaffold10x_1853_pilon-snap-gene-0.13 (Uncharacterised protein); 4: maker-scaffold10x_157_pilon-snap-gene-0.196 (EGF-like protein); 5: maker-scaffold10x_157_pilon-snap-gene-0.179 (Putative multidrug resistance protein 1, 2, 3 (P glycoprotein 1, 2, 3); ATP binding cassette subfamily B MDR TAP); 6: maker-scaffold10x_157_pilon-snap-gene-0.180 (SANT/Myb-like DNA-binding domain protein); 7: maker-scaffold10x_157_pilon-snap-gene-0.197 (ADP-ribosylation factor 2); 8: maker-scaffold10x_157_pilon-snap-gene-0.181 (RNA-binding protein sym-2/ Heterogeneous nuclear ribonucleoprotein); 9: maker-scaffold10x_157_pilon-snap-gene-0.198 (DNA directed RNA Polymerase I and III (A/C) shared subunit); 10: maker-scaffold10x_157_pilon-snap-gene-0.182 (Ras-related protein Rap-1); 11: maker-scaffold10x_157_pilon-snap-gene-0.183 (Receptor protein serine/threonine kinase); 12: maker-scaffold10x_157_pilon-augustus-gene-0.97 (D-amino-acid oxidase/ D-aspartate oxidase); 13: maker-scaffold10x_157_pilon-snap-gene-0.184 (Max-like protein X); 14: maker-scaffold10x_157_pilon-snap-gene-0.185 (EGF-like protein); 15: maker-scaffold10x_157_pilon-snap-gene-0.186 (Surfeit locus protein 4); 16: augustus_masked-scaffold10x_157_pilon-processed-gene-0.14 (TFIIH basal transcription factor complex helicase XPD subunit); 17: maker-scaffold10x_157_pilon-snap-gene-0.187 (Fatty acid binding protein V); 18: maker-scaffold10x_157_pilon-snap-gene-0.200 (Stomatin-2 / SPFH Domain / Band 7 family protein); 19: maker-scaffold10x_157_pilon-snap-gene-0.201 (Glycosylphosphatidylinositol (GPI) ethanolamine phosphate transferase 1); 20: maker-scaffold10x_157_pilon-pred_gff_StringTie-gene-0.138 (Sugar phosphate exchanger 3); 21: maker-scaffold10x_157_pilon-snap-gene-0.203 (Ribonuclease 3); 22: maker-scaffold10x_157_pilon-snap-gene-0.188 (Putative serine-rich repeat protein); 23: maker-scaffold10x_157_pilon-snap-gene-0.204 (Putative transferase CAF17, mitochondrial); 24: maker-scaffold10x_157_pilon-snap-gene-0.205 (Lamin-1/ Neurofilament protein); 25: maker-scaffold10x_157_pilon-snap-gene-0.189 (Gyf domain protein); 26: snap_masked-scaffold10x_157_pilon-processed-gene-0.72 (Prominin); 27: maker-scaffold10x_157_pilon-snap-gene-0.206 (Phospholipid transport protein / CRAL-TRIO / SEC14-like); 28: maker-scaffold10x_157_pilon-snap-gene-0.190 (Ubiquitin carboxyl-terminal hydrolase); 29: maker-scaffold10x_157_pilon-snap-gene-0.207 (Ubiquitin carboxyl-terminal hydrolase); 30: maker-scaffold10x_157_pilon-augustus-gene-0.89 (Ubiquitin carboxyl-terminal hydrolase).

advantage of bulk segregant analysis in our field isolate was that we exploited natural recombination in wildtype populations. The fact that three replicates of relatively small numbers of eggs (500) pre- and post-TCBZ treatment was sufficient to detect signals of selection raises the exciting prospect of conducting similar studies for TCBZ selection in *F. hepatica* field populations in other geographical locations and provides a valuable approach for the study of flukicide resistance more broadly, e.g. for drugs such as closantel and albendazole. Our work was conducted with adult parasites and clearly shows TCBZ resistance is a heritable trait. One of the most important aspects of our classical linkage mapping is that it allowed us to determine that, in contrast to oxamniquine and praziquantel resistance in schistosomes, TCBZ resistance is a dominant trait [24,33,34]. This tells us that once resistance emerges or is introduced within liver fluke populations it has the potential to spread rapidly [35], and highlights the need for rapid detection and effective treatment to mitigate the impact of TCBZ-R liver fluke infections in livestock and humans.

Linkage mapping has proved highly successful for identifying genetic determinants for phenotypic traits such as pathogenicity, host specificity and drug resistance in parasites of humans e.g. protozoa [36–40] and *Schistosoma* spp. [21]. More recently, population genomic analyses mapped a QTL for ivermectin resistance in the ruminant nematode, *H. contortus* [25]. Mapping studies in these parasites benefited from fully assembled genomes, something we are yet to achieve for *F. hepatica*, although recent publication of a chromosomal-level genome assembly for its sister species *F. gigantica* is encouraging [41]. Genomic resources for *F. hepatica* extend to two independent assemblies [42–44]. Our success at generating the first genetic cross and subsequent linkage mapping of drug resistance loci paves the way for studies on important phenotypic traits for *F. hepatica* in the future. The technical challenges presented by a fragmented genome have been highlighted elsewhere [25], and whilst we overcame many of these by our experimental cross and additional linkage experiments, a chromosome-level assembly will be a vital resource to progress future studies.

## Genetic mapping pinpoints candidate genes conferring triclabendazole resistance

Anthelmintic resistance can occur due to increased efflux, enhanced metabolism and through efficient detoxification mechanisms. By integrating our mapping studies and our genomic and genetic data with genes previously implicated in TCBZ resistance or TCBZ mode of action, we can prioritise specific genes that may play a role. Our mapping studies have highlighted that any of 30 genes could be involved in TCBZ resistance, but strongest selection was placed on the region where ABCB1, RAS-RP, ARF and a few other genes cluster.

ABCB1 (P-glycoprotein, Pgp), also known as MDR1, is implicated in drug resistance in multiple organisms. Overexpression of Pgp transporters, leading to increased drug efflux has been proposed as a potential route to drug resistance [45,46], and the observation that we are dealing with a dominant trait is consistent with a role for over expression of ABC transporters. In *F. hepatica* ABCB1 (Pgp)-linked drug efflux pumps have been the focus of altered drug uptake studies. Existing evidence of a role for Pgp in TCBZ resistance includes a) lower uptake of TCBZ and its metabolite TBCZ.SO in TCBZ-R compared to TCBZ-S flukes [47,48], b) reversal of the resistance phenotype *in vitro* by co-incubation with ivermectin (IVM), a known multidrug resistance (MDR) reversing agent and potential competitive substrate for Pgp [47], and c) potentiation of TCBZ action *in vitro* in TCBZ-R flukes in the presence of Pgp inhibitor R(+)-verapamil [49–51]. Therefore, our identification of ABCB1 is noteworthy, but based on our current annotation, there is no support for CNV that are thought to underlie overexpression of ABC transporters, and the lack of constitutive expression of this ABCB1 on scaffold 157 in adult parasites is inconsistent with a role in TCBZ resistance.

Specific mutations have been reported to drive transcriptional changes in Pgp genes in humans [52]. Although one SNP, $T^{687}G$, in an ABC gene from small numbers of *F. hepatica*, was originally implicated in TCBZ resistance [53], this was not supported by studies in isolates from Australia [54] or Latin America [55]. Here, we have reported two non-synonymous SNPs in the ABCB1 gene on scaffold 157 that segregate between resistant and susceptible parasites. Mining the ABCB1 gene located on scaffold 157 from multiple resistant and susceptible parasites will provide an insight into what role, if any, SNPs play in TCBZ resistance. The ability of TCBZ resistant isolates to remain susceptible to other flukicides such as closantel and albendazole [56] does not suggest a generalised role for ABCB1 in multidrug resistance but implies a level of specificity for ABCB1 on scaffold 157 in TCBZ resistance. It is worth noting that at least 11 ABCB genes have been identified in *F. hepatica*, which are located on different scaffolds [43, A. Maule, personal communication].

FABPs are small proteins that can bind anthelmintics [57] and they have been shown to be involved in drug storage and sequestration [58]. Upregulation of FABP mRNA was noted when drug resistant *Anopheles gambiae* were exposed to permethrin [59]. FABPs are known to be present in the tegument of *Fasciola* spp. [60–62]. In 2016, a systems approach revealed a FABP superfamily of seven clades, including the novel identification of the FABPV family, a representative of which is within our genomic locus [63]. The FABPV is closely related to FABP isoforms I-III [63]. Previous proteomic comparisons showed reduction in FABP synthesis (encoded by three FABP genes distinct from the FABPV gene located on scaffold 157) in a susceptible isolate exposed to TCBZ [64]. Moreover, a type I FABP Fh15 with the capacity for sequestration showed increased expression in resistant adult flukes exposed to TCBZ [64].

Ras-RP and ARF have not been implicated in TCBZ resistance based on previous candidate gene studies but are key regulators of important biological processes. The presence of a classical Ras gene, (Ras-RP) and another Ras superfamily member, ARF, within the major locus associated with TCBZ resistance is of interest for several reasons. A subfamily of Ras genes, Rabs, are small GTPases that have been linked to drug resistance in the protozoan parasite, *Leishmania donovani* [65]. In yeast, TCBZ has been shown to inhibit the production of cAMP by either direct inhibition of adenylate cyclase or by acting on the GTP-Ras-adenylyl cyclase pathway [66]. *Fasciola hepatica* adenylate cyclase is amongst the most active of any organism, its activity is thought to regulate carbohydrate metabolism and motility of the worms [67]. Adenylate cyclase in *F. hepatica* is activated by serotonin receptors that function through GTP-dependent transmembrane signalling pathways [68–71] and was identified as a potential therapeutic target in *F. hepatica* several decades ago [72]. Experiments with liver fluke tissue revealed that an endogenous ADP-ribosylation enzyme and its protein substrate were present and capable of regulating adenylate cyclase activity [73]. Our observation of a $C^{167}Y$ variant in ARF (maker-scaffold10x_157_pilon-snap-gene-0.197), that segregated in the genetic crosses and was conserved in other TCBZ-R adult fluke isolates (*Fh*LivR2, *Fh*LivR3, *Fh*LivR4*pop*) is of particular interest and warrants further investigation.

Whilst these analyses may help us narrow down which gene might be responsible drug resistance mechanisms are not restricted to mutations in coding regions and changes in gene expression. It is possible that any one of the 30 genes within the locus is responsible for resistance and, given that RNAi has been optimised for *F. hepatica* [74], systematic knockdown of each candidate gene would be a sensible way forward. RNAi, combined with recent advances in the culture of juvenile parasites [75] and *in vitro* phenotyping for TCBZ resistance, offers a powerful platform with which to screen for the causal gene, and provides opportunity to investigate whether resistance mechanisms are stage specific. Interrogation of *in vivo* RNA-seq datasets from isolates of known phenotype would inform on whether any candidate genes show differential expression on TCBZ exposure. Similarly, as small non-coding microRNA (miRNA) are known to regulate gene expression it would be prudent to look for predicted miRNA binding sites in candidate genes, particularly given that they have been linked to drug resistance in nematodes [76] and a large dataset of miRNAs has been reported in *F. hepatica* [77].

Undoubtedly one or more of these approaches will allow us to pinpoint the causal gene, but the question remains as to whether this underlying mechanism explains all observations of phenotypic resistance. We note that the signal of selection in the experimental cross and field data are adjacent rather than coincident. Out of necessity the experimental and field data used different sets of SNPs so this could be a statistical artefact. Alternatively, it may be a biological effect, indicating that different mutations circulate in the field that target the same genetic locus. This raises the interesting possibility that resistance can evolve multiple times but is constrained in the number of genome targets that can confer resistance. With the genomic

mapping approaches optimised here we now have the tools to address these more complex questions about TCBZ resistance in *F. hepatica*.

## Gene content in the major locus does not support a direct role for many gene families formerly identified as candidates for triclabendazole

In the absence of a genome-wide approach, TCBZ resistance studies relied on assumptions about involvement of candidate genes or gene families [28]. Across the genome we identified multiple candidate genes: 14 tubulin genes, 25 ABC transporter genes, three CYP450 (-like) genes, seven FABP genes, 11 glutathione S-transferase genes and three thioredoxin peroxidase (-like) genes in the *F. hepatica* genome (S7 Table). Most of these genes were located in scaffolds that showed no evidence of being under selection in our experiment, with none of the moving windows appearing in the top 1%. Furthermore, only genes on scaffold 157 showed evidence of being under selection in both our experimental and field data (S7 Table). Amongst prime candidate genes that can be excluded based on their absence within the major locus of the populations studied here are β-tubulin, the microtubule fraction known to cause BZ resistance in nematodes [78]. Although they were initially implicated in TCBZ resistance based on changes typical of microtubule inhibition in TCBZ-S but not -R flukes, [reviewed by 79–81] no differences in β-tubulin isotypes sequences or expression levels were reported between TCBZ-S and TCBZ-R flukes [64,82,83]. Whilst it may still be the case that TCBZ acts via β-tubulin, the lack of a β-tubulin gene in our mapped region rules this gene out as a candidate for directly conferring TCBZ resistance. Similarly, it has been shown that drug metabolism is upregulated in TCBZ-R flukes [48,84] possibly involving FMO, CYP450 or GST, the mu type, specifically [85–87], and an amino acid substitution $T^{143}S$ of GST in the TCBZ-R flukes has been reported [88]. The absence of GSTs, FMO or CYP450 from the locus excludes the direct action of these molecules in TCBZ resistance at least in the populations studied here. It is important to note that the mapping approach was taken with populations within a restricted geographic region of the UK and it may be that resistance is driven by different processes in other locations. Our work provides the first means with which to address whether a common mechanism of resistance occurs in *F. hepatica* populations.

## Conclusion

TCBZ is the drug of choice to treat fasciolosis in sheep and cattle, and is the only drug licenced to treat humans. Identifying genetic determinants for resistance, as we have here, is invaluable to our understanding of the mechanisms behind TCBZ resistance and how we might best mitigate its impact. In this study, we exploited the biological process of clonal expansion within the snail intermediate host and recent advances in large sequencing datasets for *F. hepatica* to further our understanding of the genetic mechanisms involved in TCBZ resistance. We have shown 1) that TCBZ resistance is primarily a single locus trait that shows dominant inheritance; 2) we have performed the first experimental genetic cross and linkage mapping study for any phenotypic trait in *F. hepatica*; 3) we successfully applied bulk segregant analysis of eggs pre- and post-treatment to detect signatures of selection within field isolates of *F. hepatica* and 4) we have conducted the first genome-wide analysis of TCBZ resistance. We have identified a small number of genes involved in membrane transport, (e.g. ABCB1), transmembrane signalling and signal transduction (e.g. Ras-RP, ARF and EGF-like proteins), DNA/RNA binding and transcriptional regulation (SANT/Myb-like DNA-binding domain protein) and drug storage and sequestration (e.g. FABP) as prime candidates for conferring TCBZ resistance. Detecting a signal of selection in naturally infected, live animals in the field provides a blueprint to determine if a common mechanism of TCBZ resistance is adopted by demographically

distinct *F. hepatica* populations and paves the way for molecular tests to detect drug resistant parasites and more effectively target treatments in both livestock and humans.

## Methods

### Ethics statement

All applicable institutional, national, and international guidelines for the care and use of animals were followed. Experimental infection in sheep was conducted under Home Office Licence PPL 40/3621 and PE77BFD98 in accordance with Animal (Scientific Procedures) Act 1986 and ethical approval for the field study was provided by the University of Liverpool Veterinary Research Ethics Committee (VREC582).

### Enhanced assembly and annotation of the *Fasciola hepatica* genome

The published assembly of *F. hepatica* [43] was improved using a combination of Hi-C and linked read data. High molecular weight DNA was prepared from *Fh*LivS1 adult fluke [32] using either Genomic-tip (Qiagen, UK) or an adapted lithium chloride and Triton X-100 lysis and phenol-chloroform extraction [89,90]. Hi-C libraries were prepared by Dovetail Genomics (Santa Cruz, CA, USA) to generate 174 million Illumina paired-end reads and scaffolded using Hi-Rise [91]. Further scaffolding was performed using linked reads from a 10X Chromium platform using Illumina reads that mapped to within 20kbp of the end of a scaffold. Scaffolds were joined where they exhibited at least 10 linked reads connecting a pair of scaffolds and where the number of links between a pair of scaffolds was at least twice as many as the next best connection. Gaps within scaffolds were then filled where possible using 2x250 bp reads [43] assembled into contigs with Discovar [92], followed by further gap filling and polishing using Illumina 2x100bp and 2x250bp reads with Pilon [93].

Annotation was performed using MAKER2 [94]. RNA-seq data [43] were used to provide initial transcript predictions by running BRAKER [95] and StringTie [96,97] to generate low quality transcript predictions and (from BRAKER) to train AUGUSTUS [95]. RepeatMasker (http://www.repeatmasker.org) was used to identify repeat regions. SNAP [98] was trained in three iterative runs of MAKER2 [94]. The completeness of the set of predicted proteins was assessed using BUSCO v3 [99] against its set of Eukaryota reference proteins and compared with the predicted proteins from *Schistosoma mansoni* (WBPS10; PRJEA36577).

### Pooled genotyping of phenotyped adult F2 populations derived from an experimental cross

The genetic crossing of a clonal TCBZ-R and -S isolate was carried out using the *Fh*LivR1 and *Fh*LivS1 isolates. Provenance, validation of phenotype and microsatellite genotyping of these two isolates was described previously [32]. The genetic cross experimental approach is shown in Fig 1. When generating F2 populations this approach required selection of F1 parasites from mating events between the two parental isolates, rather than mating between parasites of the same genotype, or self-fertilization. This was done by generating multiple single F1miracidium:snail infections of our laboratory-maintained *Galba truncatula* and screening F1 metacercariae by genotyping them for the presence of both parental multilocus genotypes [32,100]. In Experiment 1, two pools of adult flukes derived from a common population of F2 recombinants were generated by *in vivo* phenotyping in 10 sheep. This gave rise to one pool (F2 TCBZ-), comprised of a mixture of TCBZ-S and -R flukes (from untreated sheep, n = 5) and another pool (F2 TCBZ+), comprised of only TCBZ-R flukes (from treated sheep, n = 5). This process was repeated in Experiment 2, using a second, common, pool of F2 eggs (Fig 1). Each

sheep was considered a replicate, with a pool of F2 DNA generated from a known number of adult parasites from each sheep; for Experiment 1 (n = 10 sheep), TCBZ-, 52, 52, 63, 65 and 50 flukes for TCBZ+ 19, 26, 37, 40 and 27 flukes and for Experiment 2 (n = 8 sheep), TCBZ- 35, 32, 64 and 67 flukes and TCBZ+ 27, 31, 33 and 21 flukes (S1 and S2 Tables).

High quality SNPs were identified from whole-genome resequencing of five isolates [43] using Bowtie2 [101] under sensitive settings and GATK [102]. SNPs were filtered to select high confidence SNPs (i.e. those that segregated within parental isolates *Fh*LivR1 and *Fh*LivS1 and F2 progeny [32,43,100], had a quality score >100 and a depth of between 6 and 50 for each isolate. Only biallelic SNPs were used. Following discovery and filtering we identified a panel of ~9M SNPs (SNP panel 1). Illumina TruSeq libraries were generated from DNA from each F2 pool and sequenced with 2x125bp reads on an Illumina HiSeq2000. Illumina adapter was removed using Cutadapt v1.2.1 [103] and reads further trimmed with a minimum window quality score of 20 with Sickle v1.200 (https://github.com/najoshi/sickle). The counts for reference and alternate alleles in each F2 pool were generated using SAM Tools mpileup [104] and filtered to retain SNPs with coverage depth within the 10% and 90% quantiles (~7.7M SNPs). Generalised linear models (GLM) with a binomial error distribution were calculated for each SNP in R (https://www.R-project.org/) for each of the two experiments. Moving windows (containing 1000 SNPs and advanced by 100 SNPs) were calculated to give the median log-likelihood ratio (LRT) statistic associated with the difference in allele frequency between parasites in F2 TCBZ+ and F2 TCBZ- pools. Windows having a median LRT in the upper 1% quantile within each experiment were identified, and only those windows exhibiting a median LRT in the top 1% for both experiments 1 and 2 were taken forward for further analysis.

## Pooled genotyping of eggs pre and post triclabendazole treatment from a field population of *Fasciola hepatica*

Our *F. hepatica* faecal egg count reduction test (FECRT) provided a common pool of eggs from three replicate groups of 10 sheep, with the same 10 sheep sampled pre- and 21days post-TCBZ treatment [105]. Field Isolate 1 (Cumbria, UK) had a total pre-treatment egg count of 15817 (4052, 2971 and 8794 across the three groups) and the total post-treatment egg count was 3187 (1037, 978 and 1172 across the three groups). This equates to an 80% reduction and indicates the presence of treatment failure. Five hundred eggs were collected from each of the six samples and washed five times in 1ml of ddH$_2$O before being used for DNA extraction.

Given the genetic diversity inherent in fluke populations it was necessary to increase our panel of high-quality SNPs, by including SNPs previously identified in *Fh*LivSP, *Fh*LivR2, and *Fh*LivR3 [32,43], and by resequencing the genome of six individual *F. hepatica* from isolate *Fh*LivR4*pop*, a TCBZ-R *F. hepatica* population from South Wales, UK. SNPs were identified using BowTie2 [101] under sensitive settings and GATK [102] and filtered based on a quality score greater than 100 and a depth of between 6 and 50 for each isolate. This provided a ~21M SNP panel (SNP panel 2). Sequencing of eggs was performed by the Centre for Genomic Research, University of Liverpool using the NovaSeq S2 Flowcell (Illumina). The GLM procedure described above was used to compare SNPs (~14M) from pre-treatment (eggs obtained from Day 0 faecal samples) and post-treatment (egg obtained from Day 21 faecal samples).

## Linkage analysis of scaffolds under selection

We genotyped individual parasites: 249 x F2 TCBZ+ and 485 x F2 TCBZ- parasites, 45 x F1 parasites and ten parental (*Fh*LivR1 and *Fh*LivS1) parasites (S3 Table). To determine if the scaffolds under selection are linked, we analysed genotypes from 485 x F2 TCBZ- (untreated) parasites.

Genotyping was performed on a subset of 48 SNPs, from scaffolds under selection, and 16 SNPs, from scaffolds of comparable size not under selection (neutral scaffolds), were selected from SNP panel 1 and further filtered for coverage depth within the 20% and 80% quantiles. There was a preference for SNPs in exons, they were selected along the entire length of the scaffold, and they had 50bp of conserved sequence either side, to allow primer design (S4 Table). Assay design and genotyping was conducted by LGC Genomics (Hertfordshire, UK) using KASP genotyping chemistry. It was not possible to design assays for nine SNPs and after genotyping three SNPs (13_5, 13_6 and 917_3) showed monomorphic results and were not included in subsequent analyses (S4 Table).

PHASE 2.1.1 [30,31] was used to infer haplotypes from SNP data. After an initial analysis, scaffolds under selection were orientated to minimise recombinant events and PHASE was rerun with a 95% confidence cut-off, 1000 iterations, thinning interval of 10 and burn-in of 100. Haplotypes of the neutral scaffolds were inferred separately to those under selection and run with the same parameters. Arlequin 3.5.1.3 [106] was used to assess linkage disequilibrium and calculate |D'| values between all pairs of SNPs; each genotype was represented once to avoid duplication of genotypes from clones. The number of steps in the Markov chain was 100000 and the number of dememorization steps (burn-in) was 5000. False discovery rate correction [107] was used to correct *p*-values in R 3.0.1 (https://www.R-project.org/), a significance level of $p < 0.05$ was used. The R package ggplot2 was used to plot results.

## Inheritance patterns (segregation) and finer scale mapping of triclabendazole resistance genes

To track segregation and association of SNPs from parental haplotypes to recombinant F2s, numbers of haplotypes across the region under selection and for each scaffold were identified in control (TCBZ-) and treated (TCBZ+) animals and assigned to a parental genotype. We used these to determine whether one or both parental haplotypes could confer resistance and whether resistance was a dominant or recessive trait. To further localise the region associated with resistance, recombinant haplotypes were identified and recombination between SNPs used to delineate a region always inherited by parasites that survived TCBZ treatment (i.e. resistant parasites).

## Annotation of genes in region of genome under selection

The protein sequence of genes under selection (candidate genes) were run through UniProt Blast using the UniRef50 and UniProtKB_RefProtSwissProt databases [108], WormBase ParaSite Version WBPS14 (WS271) Blast against all species in the protein database [43,44,109,110] and OrthoDB version 9 against the Metazoan database [111] to determine an appropriate description and function for each candidate gene. InterPro [112] was used to identify domains as a confirmation of the protein function. WormBase ParaSite Version WBPS16 (WS279; [109,110] was used to identify orthologs in *Fasciola gigantica* (BioProject PRJNA230515; [42]), *Clonorchis sinensis* (BioProject PRJNA386618- Cs-k2;[113]), *Opisthorchis viverrini* (BioProject PRJNA222628; [114]), and *Schistosoma mansoni* (BioProject PRJEA36577; [115]). Comparative analysis was carried out against the re-assembled/re-annotated *Fasciola hepatica* genome (WormBase ParaSite version WBPS17 BioProject PRJNA179522; [42]). Differential gene expression analysis was carried out using the *F. hepatica* life cycle stage specific transcriptome datasets (ERP006566; [43]) that have been mapped to the *F. hepatica* genome assembly at WormBase ParaSite (PRJEB25283). Transcript expression values were expressed as transcripts per million (TPM) to allow for comparison between life cycle stages. Using SNPs from SNP panel 1 and 2 non-synonymous amino acid changes were identified within exons of genes 3 to

10 (Table 5). The focus was non-synonymous changes in segregating resistance alleles that were conserved in related parasites (*Fh*LivR2, *Fh*LivR3, *Fh*LivR4*pop*).

## Annotation of previously identified candidate genes

We interrogated the *F. hepatica* genome for candidate genes that had been the focus of previous TCBZ resistance studies [28]. Tubulin proteins were as previously characterised [116] and a list of ABC transporters was provided courtesy of A. Maule, personal communication (S7 Table). NCBI nucleotide and protein databases were searched for *F. hepatica* and either (i) cytochrome P450, (ii) fatty acid binding protein, (iii) glutathione S-transferase, or (iv) thioredoxin peroxidase. WormBase ParaSite Blast [109,110] was used to identify candidate genes within the *F. hepatica* genome (DNA and protein database of BioProject PRJEB25283). Genes were not included where the protein was only structurally related to the functional annotation, or only contained domains related to gene function.

## *In vivo* experimental infections

Sheep infections were carried out essentially as described previously [32]. Briefly, >12 week-old Lleyn cross lambs were infected by oral administration of ~200 (parental clones and F1) or ~400 (F2) metacercariae per sheep. Infection status was monitored weekly by ELISA [117] from four weeks prior to infection and by faecal egg count (FEC) prior to infection and from eight weeks post infection. Treated sheep were dosed orally with TCBZ (Fasinex, Novartis), at the recommended dose rate of 10 mg/kg. At 12–16 weeks post infection, sheep were humanely euthanised and enumeration of adult liver flukes was performed by dissection of the bile ducts and incubation of the liver in PBS for 2 h at 37˚C. Adult parasites manually recovered by dissection from the bile ducts, were washed in PBS, snap frozen and stored at -80˚C. Eggs for downstream infection of snails (F1 and F2 eggs) were harvested from adult parasites purged by incubation in 1–2 ml of Dulbecco's Modified Eagle's Media (DMEM; Sigma-Aldrich, Dorset, UK) containing 1000 units penicillin, 0.1 mg streptomycin and 0.25µg amphotericin B (Sigma-Aldrich, UK) for a minimum of 2 h at 37˚C.

## DNA isolation

For F2 pooled genotyping, genomic DNA was extracted from ~20mg at the anterior end of each adult fluke using the DNeasy Blood and Tissue kit (Qiagen, UK) with elution in 100µl of buffer AE. This was followed by precipitation using 3M NaOAc and isopropanol at 4˚C. Each individual fluke DNA was checked for quality on a 2% agarose gel and quantified by Quant-IT PicoGreen (Life technologies/ThermoFisher Scientific). Equimolar concentrations of genomic DNA from each parasite was mixed and purified with GenomicTip (Qiagen, UK) to create an F2 pool of high molecular weight DNA per sheep, for sequencing. Egg DNA from field isolates was extracted using the DNeasy Blood and Tissue Kit (Qiagen, UK) with the following modifications: (i) a micropestle (Argos Technologies, USA) and Pellet Pestle Motor (Kontes) were used to homogenise the eggs; (ii) RNase A was used; (iii) elution was in 100µl of buffer AE. 5µl of egg DNA was subjected to whole genome amplification using a REPL-g Mini Kit (Qiagen, UK), followed by purification using QiaAmp Mini Column (Qiagen, UK) with the following modifications (i) the DNA was added to the spin column and then the wash steps were performed (ii) elution was with 65µl of buffer AE.

## Maintenance of Fasciola hepatica in Galba truncatula

*Galba truncatula* snail stocks were maintained as described previously [32]. Briefly, they were maintained at 22˚C on pans of clay mud and fed on a diet of *Oscillatoria* spp. algae. *F. hepatica* eggs were

embryonated at 27°C for 14 days and stimulated to hatch by exposure to light. Each snail ~4 mm in height was exposed to either one miracidium (F1) or 5–8 miracidia (F2) to generate pools of clonal or multi-genotype parasites, respectively. Following infection, snails were maintained on mud, fed every 2–3 days and stimulated to shed cercariae by sealing the snail into visking tubing containing water and exposing them to a drop in temperature. The cercariae then encyst on the visking tubing as metacercariae [32]. Metacercariae from multiple snails were pooled prior to infection, to provide a dose rate of ~400 F2 parasites per sheep; 7–10 F2 metacerariae/snail (n = 41snails) for Experiment 1 and 4–10 metacerariae/snail (n = 47snails) for Experiment 2 (Fig 1).

## Supporting information

**S1 Table. Total enumeration of parasites from each animal.**
(XLSX)

**S2 Table. Number of parasites used for pooled SNP genotyping.**
(XLSX)

**S3 Table. Number of parasites used for LGC genotyping.**
(XLSX)

**S4 Table. SNPs selected for LGC genotyping.**
(XLSX)

**S5 Table. Haplotypes inferred using PHASE 2.1.1 software for scaffolds under selection.** Recombinants have been coded to match the colours of the four parental haplotypes and minimise recombination events.
(XLSX)

**S6 Table. Haplotypes inferred using PHASE 2.1.1 software for neutral scaffolds.**
(XLSX)

**S7 Table. Annotation of previously identified candidate genes.** The number of moving windows that appear in the top 1% quantile in both of the two replicate experiments and in the field data are shown for each gene.
(XLSX)

**S8 Table. Differential gene expression of the candidate genes throughout the *Fasciola hepatica* life cycle, based on average TPM values.**
(DOCX)

**S9 Table. a) Non-synonymous SNPs identified within genes 3 to 10 (Table 5) that segregate in experimental crosses; b) Non-synonymous SNPs present within genes 3 to 10 (Table 5) in post-treatment (resistant) eggs from Field Isolate 1.**
(XLSX)

**S10 Table. Results of moving windows analysis for experimental crosses used to generate Figs 2 and 5.**
(XLSX)

**S11 Table. Results of moving windows analysis for Field Isolate 1 used to generate Figs 2 and 5.**
(XLSX)

**S12 Table. Results of the linkage analysis between pairs of loci used to generate Fig 3.**
(XLSX)

## Acknowledgments

We are grateful for the advice and sequencing services provided by staff within the Centre for Genomic Research, University of Liverpool. A list of ABC transporter genes was provided courtesy of Professor Aaron Maule, Dr Erin McCammick and Dr Nathan Clarke, Queen's University Belfast. We would like to acknowledge the support of the Animal and Plant Health Agency, APHA. We would like to extend our gratitude to the farmers who provided faecal samples from sheep, and the veterinarians in private practice who collected these for us. We would like to acknowledge the help of Ms Alice Balard, Mrs Catherine Hartley, Mr Nigel Jones, Mrs Helen Smith, and Professor Rob Smith for their assistance with maintenance of snail colonies, animal care and sample collection within the Institute of Infection, Veterinary and Ecological Sciences at the University of Liverpool.

## Author Contributions

**Conceptualization:** E. James La Course, Diana J. L. Williams, Steve Paterson, Jane E. Hodgkinson.

**Data curation:** Steve Paterson.

**Formal analysis:** Nicola J. Beesley, Krystyna Cwiklinski, Terry W. Spithill, Steve Paterson, Jane E. Hodgkinson.

**Funding acquisition:** Diana J. L. Williams, Steve Paterson, Jane E. Hodgkinson.

**Investigation:** Nicola J. Beesley, Krystyna Cwiklinski, Katherine Allen, Rebecca C. Hoyle, Diana J. L. Williams, Jane E. Hodgkinson.

**Methodology:** Nicola J. Beesley, Krystyna Cwiklinski, Steve Paterson, Jane E. Hodgkinson.

**Project administration:** Jane E. Hodgkinson.

**Supervision:** Diana J. L. Williams, Steve Paterson, Jane E. Hodgkinson.

**Visualization:** Nicola J. Beesley, Krystyna Cwiklinski, Steve Paterson, Jane E. Hodgkinson.

**Writing – original draft:** Nicola J. Beesley, Steve Paterson, Jane E. Hodgkinson.

**Writing – review & editing:** Nicola J. Beesley, Krystyna Cwiklinski, Katherine Allen, Rebecca C. Hoyle, Terry W. Spithill, E. James La Course, Diana J. L. Williams, Steve Paterson, Jane E. Hodgkinson.

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
