## [Decision Letter · Decision Letter 0]

15 Nov 2022

Dear Dr Beesley,

Thank you very much for submitting your manuscript "A major locus, dominantly inherited, confers triclabendazole resistance in Fasciola hepatica" for consideration at PLOS Pathogens. As with all papers reviewed by the journal, your manuscript was reviewed by members of the editorial board and by several independent reviewers. The reviewers appreciated the attention to an important topic. Based on the reviews, we are likely to accept this manuscript for publication, providing that you modify the manuscript according to the review recommendations.

Three highly qualified reviewers find great merit in this manuscript, a conclusion with which I agree. However, Reviewer 3 (while noting that the manuscript is "outstanding') identified some issues that could be addressed by the authors to improve it. I do not think that additional experiments to attempt to further narrow the list of candidate genes are required for publication of the current data, but additional discussion as recommended (also by Reviewer 1) is warranted. I congratulate the authors on this work and look forward to receipt of a suitably revised version.

Sincerely,

Timothy G. Geary, PhD

Guest Editor

PLOS Pathogens

P'ng Loke

Section Editor

PLOS Pathogens

Kasturi Haldar

Editor-in-Chief

PLOS Pathogens

orcid.org/0000-0001-5065-158X

Michael Malim

Editor-in-Chief

PLOS Pathogens

orcid.org/0000-0002-7699-2064

Three highly qualified reviewers find great merit in this manuscript, a conclusion with which I agree. However, Reviewer 3 (while noting that the manuscript is "outstanding') identified some issues that could be addressed by the authors to improve it. I do not think that additional experiments to attempt to further narrow the list of candidate genes are required for publication of the current data, but additional discussion as recommended (also by Reviewer 1) is warranted. I congratulate the authors on this work and look forward to receipt of a suitably revised version.

Reviewer Comments (if any, and for reference):

Reviewer's Responses to Questions

**Part I - Summary**

Reviewer #1: This manuscript reports important progress in efforts to understand anthelmintic resistance, both in terms of the approaches taken and the data generated. Indeed, wrt liver fluke research, this is the most important work reported since the genome paper was published in 2015, coincidentally also by the Hodgkinson laboratory.

The manuscript charts an advance in understanding of flukicide resistance mechanisms in the liver fluke. Although the manuscript does not attempt to identify a specific resistance mechanism, it has generated the first genetic cross and linkage mapping of TCBZ resistance loci, resulting in the identification of a region encompassing ~30 genes.

Importantly, not only have the authors identified a locus that confers TCBZ resistance, but confirmed that it is also the target for drug selection in a field population. Demonstration that the resistance is a dominant trait is a very important consideration for the spread and control of TCBZ resistance and sets it apart from the few other known resistance traits in parasitic flatworms.

The link between the observations on the experimental approach and the outbred field populations is intriguing and could support an hypothesis that the genes within the locus identified here includes genes that are critical to the resistance phenotype more broadly. However, the authors indicate that the geographical proximity of the areas from where the clonal S and R isolates and the field population used in the study were recovered could explain this observation, with the rapid spread of the dominant trait within the region. Clearly, the tools are now available to interrogate flukicide resistance traits in more geographically disparate fluke populations.

The approaches taken overcame several key hurdles and were very well considered, providing much confidence in the data generated. The manuscript is very well written, indeed unusually succinct and to the point, which is refreshing.

Reviewer #2: Beesley et al present the first description of genetic loci associated/linked with the development of resistance to triclabendazole in Fasciola spp.. The rise of drug resistance to triclabendazole is of significant concern for livestock producers globally. The complex, indirect lifecycle of Fasciola hepatica, which includes asexual and sexual reproduction phases, introduces significant challenges to a traditional genome-wide association study that aims to explore the underlying genomic basis for drug resistance. The authors addressed these challenges herein and provide the first comprehensive support for resistance markers to triclabendazole. By conducting a long-term experiment to genetically cross a genetically diverse parasite, they present the first robust support for the landscape of resistance in the laboratory and field and infer that the trait is dominantly inherited. It is of no surprise to see the resistance landscape is complex, even within a small region of the UK. The resource provided herein will be an invaluable resource for exploring TCBZ resistance globally (as they discuss herein).

Reviewer #3: This paper uses genetic crosses and linkage analysis to localize the gene(s) determining triclabendazole (TCBZ) resistance to a small genome region containing 30 genes. In brief, the paper describes two generation genetic crosses, followed by bulk segregant analysis of F2 progeny populations to identify QTL regions. They then use genotyping of individual flukes to demonstrate that the QTLs identified co-segregate and are therefore close together in the genome, and to show that TCBZ resistance shows dominant inheritance. As an additional validation, they show that the same genome regions is selected during treatment of natural parasite infections. Finally, they examine the genome content of the QTL regions, showing that it contains several promising candidate loci.

I enjoyed reading this tour-de-force paper enormously. The paper has many strengths. It is carefully written in accessible language and logically structured. The experimental design is carefully designed and appropriate. The novelty and significance is extremely high, because triclabendazole (TCBZ) is the first line drug against Fasciola, as this trematode genus is naturally resistant to Praziquantel, and unbiased genetic analyses of this nature have not been applied to Fasciola. Several aspects of project design were especially commendable. Use of replicate crosses is a strong feature, and gives confidence in the robustness of the results. To achieve this work, the authors significantly improved the genome assembly for this important parasite. The authors used analysis of individual progeny parasite, to determine the relative location of scaffolds selected by TCBC treatment, and inheritance of TCBC. The results are further validated by examination of selected scaffolds in naturally infected animals after treatment with TCBZ.

The paper stops short of identifying the gene (s) that drive resistance and completing this story. The QTL contains several candidate loci of interest. I would like to see some further exploration of these candidates by examining variants segregating in the crosses - some suggestions of how this can be done are given in section II. Likewise, some further discussion of strategies for validation of causative loci would improve the paper.

**Part II – Major Issues: Key Experiments Required for Acceptance**

Reviewer #1: (No Response)

Reviewer #2: The experiments were carefully linked and the interpretation of their results was comprehensive and easy to understand

Reviewer #3: The central conclusion – that TCBZ resistance maps to a single QTL containing 30 genes, and shows dominant inheritance – is well supported from duplicate mapping experiments and additional verification in field derives infections. However, further discussion of how future work will narrow down the gene involved and the validation approaches (e.g. RNAi) that can be used should be included in this paper.

The paper stops short of identifying the gene (s) that drive resistance and completing this story. The QTL contains several candidate loci of interest. Further use of genetic data to narrow down the list of genes under this QTL could help to guide further efforts to determine the causative gene. I’d suggest some further exploration of these candidate genes. For example, genes that are not expressed in adult worms are unlikely to be involved and could be excluded. Genes that are invariant, or have only synonymous mutations can be deprioritized (with the caveat that non coding changes may also be involved), while genes carrying non-synonymous changes with the expected segregation pattern can by prioritized. I suggest including a table or supplement showing the amino acid changes segregating in crosses within candidate loci, and in related flukes for the genes 3-10 in table 4. This would be informative for readers, and helpful for determining possible causative genes. For example, if a candidate gene is conserved across related flukes, but shows non-synonymous change in segregating resistant alleles, this would be of particular interest.

Copy number variants often underlie overexpression of ABC transporters – does the genomic data provide any evidence for this? Similarly, overexpression of genes within the QTL could be used to identify prime candidates. As the number of genes is quite small (30), systematic gene knockdown experiments with RNAi could be used – this gene manipulation approach is possible for Fasciola (https://www.ncbi.nlm.nih.gov/pmc/articles/PMC5749881/) and should be mentioned. The paper would be strengthen by inclusion of a further discussion section detailing what strategies can be used to validate the causative resistance loci and variants.

An exciting (and robust) finding of this paper is that TCBZ resistance shows dominant inheritance. In contrast loci that impact response to oxamniquine and praziquantel in another fluke (S. mansoni) show recessive inheritance, suggesting loss-of-function. Dominant inheritance may suggests overexpression of allelic variants underlies resistance – for example overexpression of ABC transporters that remove drug from parasite cells might be expected to show dominant or co-dominant expression. Some discussion of what dominant inheritance suggests about the mechanism of resistance could also be useful.

Methodology clarification needed: Beesley et al find that several different scaffolds contains markers linked to TCBZ resistance. They then genotype individual F2s to determine whether these scaffolds are close together (and indicate a single locus) or distant, suggesting >1 unlinked loci underlie resistance. This is a very sensible and elegant approach, and provided strong evidence that these scaffolds are closely linked in the genome. However it is not clear exactly how this analysis was done.

Ln 202-204 states: “We genotyped 249 x F2 TCBZ+ and 485 x F2 TCBZ- parasites (S3 Table) with a subset of 48 SNPs derived from each of the six mapped scaffolds and 16 SNPs from neutral (not under selection) scaffolds of comparable size” It would be best to do this analysis with only 485 x F2 TCBZ- (untreated) parasites, as this will allow them to investigate segregation without drug selection.

**Part III – Minor Issues: Editorial and Data Presentation Modifications**

Reviewer #1: Some minor comments for attention of authors:

The segment in the discussion that summarises the main data implicating pgps in flukicide resistance needs some attention. For example, under (b) the context for the role of ivermectin (potentially as a competitive substrate for pgp activity) will not be clear for the reader and so needs some explanation. Further, the third piece of evidence under (c) is incomplete, appearing to be accidentally truncated at pgp – the authors may have meant the pgp inhibitor verapamil(R-)?

On lines 388 and 392, where genes previously implicated in TCBZ resistance are being ruled out because of their absence from the resistance loci, the authors should consider defining that these genes are being ruled out as the basis for TCBZ resistance in the fluke clonal and field populations studied within the MS.

L414 ‘ as prime candidates for conferring’….

To be considered by authors, but not essential to publication:

Amongst the 30 genes identified, the potential role of the ABCB1 gene in TCBZ resistance is intriguing. Notably, a cursory glance at its expression in liver fluke transcriptomic datasets suggests very low (or no) expression in juvenile liver fluke. This could diminish the likelihood of that specific gene being a key player, assuming that the resistance phenotype in question is not confined to the adult worm. The clonal resistant isolates used in the study were exposed to TCBZ treatments during patent infections, but it is unclear if there are data on the susceptibility of the juveniles? If the authors know that the resistant trait is present in both the adult and juvenile worms (or indeed is restricted to the adult), this could provide impetus for looking at stage-specific expression patterns of the 30 ‘candidate’ genes to see if any are expressed in line with the resistance phenotype.

Did the authors consider if the locus highlighted in the study contained any non-coding genes and/or any known binding sites for known miRNAs? If not this could be highlighted as something for consideration in future work.

Reviewer #2: The explanation of their methods/approach was comprehensive but easy to understand and the authors have made all their findings freely available via supplementary materials or via online databases. It is a shame that they were unable to complete a reference-level genome for their analysis but it wasn’t from the lack of trying. The challenges with achieving chromosome-level genomes for large eukaryote genomes with long-read and short DNA libraries alone is well known.

The manuscript was extremely well written and I don’t have any comments to be addressed regarding format of style.

Reviewer #3: Ln 58. “Our study is the first of its kind for liver fluke.” I suggest removing this statement – this is not needed to sell the science. Likewise, I’d remove “We have delivered several firsts” (Conclusion – ln 405).

Ln 83-83: “As a result, enhanced molecular surveillance of resistance allele frequencies in field populations has facilitated targeted treatments with efficacious drugs and reduced selection pressure for resistance development [22]”

This is somewhat overstated – the discovery of SmSULT-OR has allowed molecular surveillance for OXA-R alleles in natural populations, but this is of limited utility from a practical standpoint, because OXA is no longer used to treat schistosome infections. I’d change to “This has allowed global mapping of OXA-R alleles in natural populations” and you could also cite https://pubmed.ncbi.nlm.nih.gov/31652296/

It is strange that several prior candidate genes all fall in the QTL region. My expectation is that just one of these genes is driving the QTL. This would be worth addressing in the discussion. See ln 256-261 “…namely an ADP ribosylation factor (Gene 7: ARF, maker256 scaffold10x_157_pilon-snap-gene-0.197; Fig. 5; Table 4), a Ras-related protein (Gene 10: 257 Ras-RP, maker-scaffold10x_157_pilon-snap-gene-0.182; Fig. 5; Table 4), an ABCB1 gene 258 (Gene 5: ABCB1, maker-scaffold10x_157_pilon-snap-gene-0.179; Fig. 5; Table 4). Lying 259 within the mapped locus, albeit slightly outside the strongest signal of selection, is another 260 gene that has been the focus of candidate gene studies for TCBZ resistance, a FABP V gene 261 (Gene 17: FABPV, maker-scaffold10x_157_pilon-snap-gene-0.187; Fig. 5; Table 4).”

Ln 613: “and stimulated to shed cercariae onto visking tubing” . Please explain or add reference for the non-Fasciola audience.

PLOS authors have the option to publish the peer review history of their article (what does this mean?). If published, this will include your full peer review and any attached files.

Reviewer #1: **Yes: **Aaron Maule

Reviewer #2: No

Reviewer #3: No

Figure Files:

Data Requirements:

Reproducibility:

References:

---

## [Editor Report · Decision Letter 1]

22 Dec 2022

Dear Dr Beesley,

We are pleased to inform you that your manuscript 'A major locus confers triclabendazole resistance in *Fasciola hepatica* and shows dominant inheritance' has been provisionally accepted for publication in PLOS Pathogens.**

*Before your manuscript can be formally accepted you will need to complete some formatting changes, which you will receive in a follow up email. A member of our team will be in touch with a set of requests.*

*Please note that your manuscript will not be scheduled for publication until you have made the required changes, so a swift response is appreciated.*

*IMPORTANT: The editorial review process is now complete. PLOS will only permit corrections to spelling, formatting or significant scientific errors from this point onwards. Requests for major changes, or any which affect the scientific understanding of your work, will cause delays to the publication date of your manuscript.*

*Should you, your institution's press office or the journal office choose to press release your paper, you will automatically be opted out of early publication. We ask that you notify us now if you or your institution is planning to press release the article. All press must be co-ordinated with PLOS.*

*Thank you again for supporting Open Access publishing; we are looking forward to publishing your work in PLOS Pathogens.*

*Best regards,*

Timothy G. Geary, PhD

Guest Editor

PLOS Pathogens

P'ng Loke

Section Editor

PLOS Pathogens

Kasturi Haldar

Editor-in-Chief

PLOS Pathogens

orcid.org/0000-0001-5065-158X

Michael Malim

Editor-in-Chief

PLOS Pathogens

orcid.org/0000-0002-7699-2064

The authors are thanked for their positive and constructive responses to the concerns raised during the review process. The new data also provide additional value to the manuscript.
---

## [Editor Report · Acceptance letter]

21 Jan 2023

Dear Dr Beesley,

We are delighted to inform you that your manuscript, "A major locus confers triclabendazole resistance in Fasciola hepatica and shows dominant inheritance," has been formally accepted for publication in PLOS Pathogens.

Best regards,

Kasturi Haldar

Editor-in-Chief

PLOS Pathogens

orcid.org/0000-0001-5065-158X

Michael Malim

Editor-in-Chief

PLOS Pathogens

orcid.org/0000-0002-7699-2064